# Efficient Data Subset Selection to Generalize Training Across Models: Transductive and Inductive Networks

**Eeshaan Jain[§], Tushar Nandy[§], Gaurav Aggarwal[†],**
**Ashish Tendulkar[†], Rishabh Iyer[*] and Abir De[§]**
[§]IIT Bombay, [†]Google, [*]UT Dallas
{eeshaan,tnandy,abir}@cse.iitb.ac.in,
{gauravaggarwal,ashishvt}@google.com, rishabh.iyer@utdallas.edu

## Abstract

Existing subset selection methods for efficient learning predominantly employ discrete combinatorial and model-specific approaches which lack generalizability. For an unseen architecture, one cannot use the subset chosen for a different model. To tackle this problem, we propose SUBSELNET, a trainable subset selection framework, that generalizes across architectures. Here, we first introduce an attention-based neural gadget that leverages the graph structure of architectures and acts as a surrogate to trained deep neural networks for quick model prediction. Then, we use these predictions to build subset samplers. This naturally provides us two variants of SUBSELNET. The first variant is transductive (called as Transductive-SUBSELNET) which computes the subset separately for each model by solving a small optimization problem. Such an optimization is still super fast, thanks to the replacement of explicit model training by the model approximator. The second variant is inductive (called as Inductive-SUBSELNET) which computes the subset using a trained subset selector, without any optimization. Our experiments show that our model outperforms several methods across several real datasets.

## 1 Introduction

In the last decade, neural networks have drastically enhanced the performance of state-of-the-art ML models. However, they often demand massive data to train, which renders them heavily contingent on the availability of high-performance computing machineries such as GPUs and RAM. Such resources entail heavy energy consumption, excessive $CO_2$ emission, and maintenance cost.

Driven by this challenge, a recent body of work focuses on suitably selecting a subset of instances so that the model can be trained quickly using lightweight computing infrastructure [4, 23, 51, 32, 54, 37, 18–21, 36]. However, these methods are not generalizable across architectures— the subset selected by such a method is tailored to train only one specific architecture and thus need not be optimal for training another architecture. Hence, to select data subsets for a new architecture, they need to be run from scratch. However, these methods rely heavily on discrete combinatorial algorithms, which impose significant barriers against scaling them for multiple unseen architectures. Appendix C contains further details about related work.

### 1.1 Our contributions

Responding to the above limitations, we develop SUBSELNET, a trainable subset selection framework. Specifically, we make the following contributions.

**Novel framework on subset selection that generalizes across models.** SUBSELNET is a subset selector that generalizes across architectures. Given a dataset, once SUBSELNET is trained on a set of model architectures, it can quickly select a small optimal training subset for any unseen (test) architecture, without any explicit training of this test model. SUBSELNET is a non-adaptive method

37th Conference on Neural Information Processing Systems (NeurIPS 2023).

since it learns to select the subset before the training starts for a new architecture, instead of adaptively selecting the subset during the training process. Our framework has several applications in the context of AutoML [35, 68, 30, 43, 61, 9, 2, 24, 22, 3]. For example, Network Architecture Search (NAS) can have a signficant speed-up when the architectures during selection can be trained on the subsets provided by our method, as compared to the entire dataset. In hyperparameter selection, such as the number and the widths of layers, learning rates or scheduler-specific hyperparameters, we can train each architecture on the corresponding data subset obtained from our method to quickly obtain the trained model for cross-validation.

**Design of neural pipeline to eschew model training for new architecture.** We initiate our investigation by writing down a combinatorial optimization problem instance that outputs a subset specifically for one given model architecture. Then, we gradually develop SUBSELNET, by building upon this setup. The key blocker in scaling up a model-specific combinatorial subset selector across different architectures is the involvement of the model parameters as optimization variables along with the candidate data subset. We design the neural pipeline of SUBSELNET to circumvent this blocker specifically. This neural pipeline consists of the following three components: **(1)** GNN-guided architecture encoder: This converts the architecture into an embedded vector space. **(2)** Neural model approximator: It approximates the predictions of a trained model for any given architecture. Thus, it provides the accuracy of a new (test) model per instance without explicitly training it. **(3)** Subset sampler: It uses the predictions from the model approximator and an instance to provide a selection score of the instance. Due to the architecture encoder and the neural approximator, we do not need to explicitly train a test model for selecting the subset since the model approximator directly provides the predictions the model will make.

**Transductive and Inductive SUBSELNET.** Depending on the functioning of the subset sampler in the final component of our neural pipeline, we design two variants of our model.

*Transductive*-SUBSELNET*:* The first variant is transductive in nature. For each new architecture, we utilize the the model approximator's predictions for replacing the model training step in the original combinatorial subset selection problem. However, the candidate subset still remains involved as an optimization variable. Thus, we still solve a fresh optimization problem with respect to the selection score provided by the subset sampler every time we encounter a new architecture. However, the direct predictions from the model approximator allow us to skip explicit model training, making this strategy extremely fast in terms of memory and time.

*Inductive*-SUBSELNET*:* In contrast to Transductive-SUBSELNET, the second variant does not require to solve any optimization problem and instead models the selection scores using a neural network. Consequently, it is extremely fast.

We compare our method against six state-of-the-art methods on five real world datasets, which show that SUBSELNET provides the best trade-off between accuracy and inference time as well as accuracy and memory usage, among all the methods.

## 2 Preliminaries

**Setting.** We are given a set of training instances $\{(\boldsymbol{x}_i, y_i)\}_{i \in D}$ where we use $D$ to index the data. Here, $\boldsymbol{x}_i \in \mathbb{R}^{d_x}$ denotes the features, and $y_i \in \mathcal{Y}$ denotes the labels. In our experiments, we consider $\mathcal{Y}$ as a set of categorical labels. However, our framework can also be used for continuous labels. We use $m$ to denote a neural architecture and represent its parameterization as $m_\theta$. We also use $\mathcal{M}$ to denote the set of neural architectures. Given an architecture $m \in \mathcal{M}$, $G_m = (V_m, E_m)$ provides the graph representation of $m$, where the nodes $u \in V_m$ represent the *operations* and the $e = (u_m, v_m)$ indicates an edge, where the output given by the operation represented by the node $u_m$ is fed to one of the operands of the operation given by the node $v_m$. Finally, we use $H(\cdot)$ to denote the entropy of a probability distribution and $\ell(m_\theta(\boldsymbol{x}), y)$ as the cross entropy loss hereafter.

### 2.1 Combinatorial subset selection for efficient learning

Given a dataset $\{(\boldsymbol{x}_i, y_i)\}_{i \in D}$ and a model architecture $m \in \mathcal{M}$ with its neural parameterization $m_\theta$, the goal of a subset selection algorithm is to select a small subset of instances $S$ with $|S| = b << |D|$ such that training $m_\theta$ on the subset $S$ gives nearly the same accuracy as training on the entire dataset $D$. Existing works [20, 47, 19] adopt different strategies to achieve this goal, but all of them aim to simultaneously optimize for the model parameters $\theta$ as well as the candidate subset $S$. At the outset,

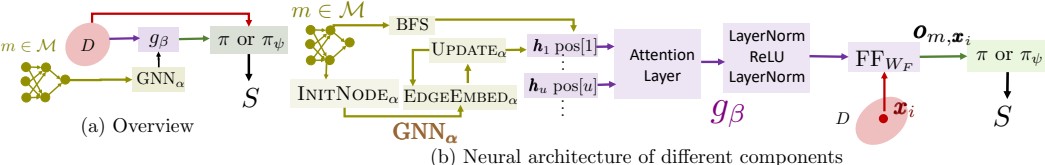

(a) Overview

(b) Neural architecture of different components

Figure 1: Illustration of SUBSELNET. (a) Overview: Given a model architecture $m \in \mathcal{M}$, SUB-SELNET takes its graph $G_m$ as input to the architecture encoder $\text{GNN}_\alpha$ to compute the architecture embedding. This, together with $\boldsymbol{x}$ is fed into the model approximator $g_\beta$ which predicts the output of the trained model $m_{\theta^*}(\boldsymbol{x})$. Then this is fed as input to the subset sampler $\pi$ to obtain the training subset $S$. (b) Neural architecture of different components: $\text{GNN}_\alpha$ consists of recursive message passing layer. The model approximator $g_\beta$ performs a BFS ordering on the emebddings $\boldsymbol{H}_m = \{\boldsymbol{h}_u\}$ and feeds them into a transformer. Subset sampler optimizes for $\pi$ either via direct optimization for $\pi$ (Transductive) or via a neural network $\pi_\psi$ (Inductive).

we may consider the following optimization problem.

$$\underset{\theta, S \subset D : |S| = b}{\text{minimize}} \sum_{i \in S} \ell(m_\theta(\boldsymbol{x}_i), y_i) - \lambda \, \text{DIVERSITY}(\{\boldsymbol{x}_i \,|\, i \in S\}), \qquad (1)$$

where $b$ is the budget, $\text{DIVERSITY}(\{\boldsymbol{x}_i \,|\, i \in S\})$ measures the representativeness of $S$ with respect to the whole dataset $D$ and $\lambda$ is a regularizing coefficient. One can use submodular functions [11, 17] like Facility Location, Graph Cut, or Log-Determinant to model $\text{DIVERSITY}(\{\boldsymbol{x}_i \,|\, i \in S\})$. Here, $\lambda$ trades off between training loss and diversity.

**Bottlenecks of the combinatorial optimization** (1)**.** For every new architecture $m$, one needs to solve a fresh version of the optimization (1) problem from scratch to find $S$. Therefore, this is not generalizable across architectures. Moreover, the involvement of both combinatorial and continuous optimization variables, prevents the underlying solver from scaling across multiple architectures.

We address these challenges by designing a neural surrogate of the objective (1), which would lead to the generalization of subset selection across different architectures.

## 3 Overview of SUBSELNET

Here, we give an outline of our proposed model SUBSELNET that leads to substituting the optimization (1) with its neural surrogate, which would enable us to compute the optimal subset $S$ for an unseen model, once trained on a set of model architectures.

### 3.1 Components

At the outset, SUBSELNET consists of three key components: (i) the architecture encoder, (ii) the neural approximator of the trained model, and (iii) the subset sampler. Figure 1 illustrates our model.

**GNN-guided encoder for neural architectures.** Generalizing any task across the different architectures requires the architectures to be embedded in vector space. Since a neural architecture is essentially a graph between multiple operations, we use a graph neural network (GNN) [59] to achieve this goal. Given a model architecture $m \in \mathcal{M}$, we first feed the underlying DAG $G_m$ into a GNN ($\text{GNN}_\alpha$) with parameters $\alpha$, which outputs the node representations for $G_m$, i.e., $\boldsymbol{H}_m = \{\boldsymbol{h}_u\}_{u \in V_m}$.

**Approximator of the trained model $m_{\theta^*}$.** To tackle lack of generalizability of the optimization (1), we design a neural model approximator $g_\beta$ which approximates the predictions of any trained model for any given architecture $m$. To this end, $g_\beta$ takes input as $\boldsymbol{H}_m$ and the instance $\boldsymbol{x}_i$ and compute $g_\beta(\boldsymbol{H}_m, \boldsymbol{x}_i) \approx m_{\theta^*}(\boldsymbol{x}_i)$. Here, $\theta^*$ is the set of learned parameters of the model $m_\theta$ on dataset $D$.

**Subset sampler.** We design a subset sampler using a probabilistic model $\text{Pr}_\pi(\bullet)$. Given a budget $b$, it sequentially draws instances $S = \{s_1, ..., s_b\}$ from a softmax distribution of the logit vector $\pi \in \mathbb{R}^{|D|}$ where $\pi(\boldsymbol{x}_i, y_i)$ indicates a score for the element $(\boldsymbol{x}_i, y_i)$. We would like to highlight that we use $S$ as an ordered set of elements, selected in a sequential manner. However, such an order does not affect the trained model, which is inherently invariant of permutations of the training data; it only affects the choice of $S$. Now, depending on how we compute $\pi$ during *test*, we have two variants of SUBSELNET: Transductive-SUBSELNET and Inductive-SUBSELNET.

*Transductive*-SUBSELNET*:* During test, since we have already trained the architecture encoder $\text{GNN}_\alpha$ and the model approximator $g_\beta$, we do not have to perform any training when we select a subset for an unseen architecture $m'$, since the trained model can then be replaced with $g_\beta(\boldsymbol{H}_{m'}, \boldsymbol{x}_i)$. Thus, the key bottleneck of solving the combinatorial optimization (1)— training the model simultaneously with exploring for $S$— is ameliorated. Now, we can perform optimization over $\pi$, each time for a new architecture. However, since no model training is involved, such explicit optimization is fast enough and memory efficient. Due to explicit optimization every time for an unseen architecture, this approach is transductive in nature.

*Inductive*-SUBSELNET*:* Here, we introduce a neural network to approximate $\pi$, which is trained together with $\text{GNN}_\alpha$ and $g_\beta$. This allows us to directly select the subset $S$ without explicitly optimizing for $\pi$, unlike Transductive-SUBSELNET.

## 3.2 Training and inference

**Training objective.** Using the approximation $g_\beta(\boldsymbol{H}_m, \boldsymbol{x}_i) \approx m_{\theta^*}(\boldsymbol{x}_i)$, we replace the combinatorial optimization problem in Eq. (1) with a continuous optimization problem, across different model architectures $m \in \mathcal{M}$. To that goal, we define

$$\Lambda(S; m; \pi, g_\beta, \text{GNN}_\alpha) = \sum_{i \in S} \ell(g_\beta(\boldsymbol{H}_m, \boldsymbol{x}_i), y_i) - \lambda H(\text{Pr}_\pi(\bullet)) \text{ with, } \boldsymbol{H}_m = \text{GNN}_\alpha(G_m) \quad (2)$$

and seek to solve the following problem:

$$\min_{\pi, \alpha, \beta} \sum_{m \in \mathcal{M}} \mathop{\mathbb{E}}_{S \sim \text{Pr}_\pi} \left[ \Lambda(S; m; \pi, g_\beta, \text{GNN}_\alpha) + \sum_{i \in S} \gamma KL(g_\beta(\boldsymbol{H}_m, \boldsymbol{x}_i), m_{\theta^*}(\boldsymbol{x}_i)) \right] \quad (3)$$

Here, we use entropy on the subset sampler $H(\text{Pr}_\pi(\bullet))$ to model the diversity of samples in the selected subset. We call our neural pipeline, which consists of architecture encoder $\text{GNN}_\alpha$, the model approximator $g_\beta$, and the subset selector $\pi$, as SUBSELNET. In the above, $\gamma$ penalizes the difference between the output of the model approximator and the prediction made by the trained model, which allows us to generalize the training of different models $m \in \mathcal{M}$ through the model $g_\beta(\boldsymbol{H}_m, \boldsymbol{x}_i)$.

# 4 Design of SUBSELNET

**Bottlenecks of end-to-end training and proposed multi-stage approach.** End-to-end optimization of the above problem is difficult for the following reasons. (i) Our architecture representation $\boldsymbol{H}_m$ only represents the architectures and thus should be independent of the parameters of the architecture $\theta$ and the instances $\boldsymbol{x}$. End-to-end training can make them sensitive to these quantities. (ii) To enable the model approximator $g_\beta$ accurately fit the output of the trained model $m_\theta$, we need explicit training for $\beta$ with the target $m_\theta$.

In our multi-stage training method, we first train the architecture encoder $\text{GNN}_\alpha$, then the model approximator $g_\beta$ and then train our subset sampler $\text{Pr}_\pi$ (resp. $\text{Pr}_{\pi_\psi}$) for the transductive (inductive) model. In the following, we describe the design and training of these components in details.

## 4.1 Design of architecture encoder using graph neural network

Architectures can be represented as directed acyclic graphs with forward message passing. During forward computation, at any layer for node $v$, the output $a(v)$ can be represented as $a(v) = \texttt{Act}\left(\sum_{u \in \text{InNbr}(v)} \texttt{Op}_v(a(u))\right)$ with the root output as the input. Here, $\texttt{Act}$ is the activation function and $\texttt{Op}_\bullet$ are operation on a node of the network. Given a GNN has a similar computation process, the permutation-equivariant node representations generated are good representations of the operations within the architecture. This allows further coupling with transformer-based architectures since they are universal approximators of permutation equivariant functions [63].

**Neural parameterization.** Given a model $m \in \mathcal{M}$, we compute the representations $\boldsymbol{H}_m = \{\boldsymbol{h}_u | u \in V_m\}$ by using a graph neural network $\text{GNN}_\alpha$ parameterized with $\alpha$, following the proposal of Yan et al. [59]. We first compute the feature vector $\boldsymbol{f}_u$ for each node $u \in V_m$ using the one-hot encoding of the associated *operation* (*e.g.*, max, sum) and then feeding it into a neural network to compute an initial node representation $\boldsymbol{h}_u[0] = \text{INITNODE}_\alpha(\boldsymbol{f}_u)$. Then, we use a message-passing network, which collects signals from the neighborhood of different nodes and recursively computes the node representations [59, 58, 12]. Given a maximum number of recursive layers $K$ and the node

$u$, we compute the node embeddings $\boldsymbol{H}_m = \{\boldsymbol{h}_u | u \in V_m\}$ by gathering information from the $k < K$ hops using $K$ recursive layers as follows.

$$\boldsymbol{h}_{(u,v)}[k] = \text{EDGEEMB}_\alpha(\boldsymbol{h}_u[k], \boldsymbol{h}_v[k]), \ \ \boldsymbol{h}_u[k+1] = \text{UPDATE}_\alpha\left(\boldsymbol{h}_u[k], \sum_{v \in \text{Nbr}(u)} \boldsymbol{h}_{(u,v)}[k]\right) \quad (4)$$

Here, $\text{Nbr}(u)$ is the set of neighbors of $u$. EDGEEMB is injective mappings, as used in [58]. Note that trainable parameters from EDGEEMB and UPDATE are decoupled. They are represented as the set of parameters $\alpha$. Finally, we obtain our node representations as $\boldsymbol{h}_u = [\boldsymbol{h}_u[0], .., \boldsymbol{h}_u[K-1]]$.

**Parameter estimation.** We perform unsupervised training of $\text{GNN}_\alpha$ using a variational graph autoencoder (VGAE). This ensures that the architecture representations $\boldsymbol{H}_m$ remain insensitive to the model parameters. We build the encoder and decoder of our GVAE by following existing works on graph VAEs [59, 46]. Given a graph $G_m$, the encoder $q(\mathcal{Z}_m \mid G_m)$, which takes the node embeddings $\{\boldsymbol{h}_u\}_{u \in V_m}$ and maps it into the latent space $\mathcal{Z}_m = \{\boldsymbol{z}_u\}_{u \in V_m}$. Specifically, we model the encoder $q(\mathcal{Z}_m \mid G_m)$ as: $q(\boldsymbol{z}_u \mid G_m) = \mathcal{N}(\mu(\boldsymbol{h}_u), \Sigma(\boldsymbol{h}_u))$. Here, both $\mu$ and $\Sigma$ are neural networks. Given a latent representation $\mathcal{Z}_m = \{\boldsymbol{z}_u\}_{u \in V_m}$, the decoder models a generative distribution of the graph $G_m$ where the presence of an edge is modeled as Bernoulli distribution $\text{BERNOULLI}(\sigma(\boldsymbol{z}_u^\top \boldsymbol{z}_v))$. Thus, we model the decoder as $p(G_m \mid \mathcal{Z}) = \prod_{(u,v) \in E_m} \sigma(\boldsymbol{z}_u^\top \boldsymbol{z}_v) \cdot \prod_{(u,v) \notin E_m}[1 - \sigma(\boldsymbol{z}_u^\top \boldsymbol{z}_v)]$. Here, $\sigma$ is a parameterized sigmoid function. Finally, we estimate $\alpha, \mu, \Sigma$, and $\sigma$ by maximizing the evidence lower bound (ELBO): $\max_{\alpha,\mu,\Sigma,\sigma} \mathbb{E}_{\mathcal{Z} \sim q(\bullet \mid G_m)}[p(G_m \mid \mathcal{Z})] - \text{KL}(q(\bullet \mid G_m) || p_{\text{pr}}(\bullet))$.

## 4.2 Design of model approximator

**Neural parameterization.** Having computed the architecture representation $\boldsymbol{H}_m = \{\boldsymbol{h}_u \mid u \in V_m\}$, we next design the model approximator, which leverages these embeddings to predict the output of the trained model $m_{\theta^*}(\boldsymbol{x}_i)$. To this aim, we developed a model approximator $g_\beta$ parameterized by $\beta$ that takes $\boldsymbol{H}_m$ and $\boldsymbol{x}_i$ as input and attempts to predict $m_{\theta^*}(\boldsymbol{x}_i)$, *i.e.*, $g_\beta(\boldsymbol{H}_m, \boldsymbol{x}_i) \approx m_{\theta^*}(\boldsymbol{x}_i)$. It consists of three steps. In the first step, we generate an order on the nodes. Next, we feed the representations $\{\boldsymbol{h}_u\}$ in this order into a self-attention-based transformer layer. Finally, we combine the output of the transformer and $\boldsymbol{x}_i$ using a feedforward network to approximate the model output.

*Node ordering using BFS order.* We first sort the nodes using breadth-first-search (BFS) order $\rho$. Similar to You et al. [62], this sorting method produces a sequence of nodes and captures subtleties like skip connections in the network structure $G_m$.

*Attention layer.* Given the BFS order $\rho$, we pass the representations $\boldsymbol{H}_m = \{\boldsymbol{h}_u \mid u \in V_m\}$ in the sequence $\rho$ through a self-attention-based transformer network. Here, the Query, Key, and Value functions are realized by linear networks on $\boldsymbol{h}_\bullet$. We compute an attention-weighted vector $\boldsymbol{\zeta}_u$ as:

$$\text{Att}_u = \boldsymbol{W}_c^\top \sum_v a_{u,v} \text{Value}(\boldsymbol{h}_v) \text{ with, } \quad a_{u,v} = \text{SOFTMAX}_v\left(\text{Query}(\boldsymbol{h}_u)^\top \text{Key}(\boldsymbol{h}_v)/\sqrt{k}\right) \quad (5)$$

Here $k$ is the dimension of the latent space, and the softmax operation is over the node $v$. Subsequently, for each node $u$, we use a feedforward network, preceded and succeeded by layer normalization operations to obtain an intermediate representation $\boldsymbol{\zeta}_u$ for each node $u$. We present additional details in Appendix D. Finally, we feed $\boldsymbol{\zeta}_u$ for the last node $u$ in the sequence $\rho$, *i.e.*, $u = \rho(|V_m|)$, along with the feature $\boldsymbol{x}_i$ into a feedforward network parameterized by $\boldsymbol{W}_F$ to model the prediction $m_{\theta^*}(\boldsymbol{x}_i)$. Thus, the final output of $g_\beta(\boldsymbol{H}_m, \boldsymbol{x}_i)$ is

$$\boldsymbol{o}_{m,\boldsymbol{x}_i} = \text{FF}_{\boldsymbol{W}_F}(\boldsymbol{\zeta}_{\rho(|V_m|)}, \boldsymbol{x}_i) \quad (6)$$

Here, $\boldsymbol{W}_\bullet$, parameters of Query, Key and Value and layer normalizations form $\beta$.

**Parameter estimation.** We train our model approximator $g_\beta$ by minimizing the KL-Divergence between the approximated prediction $g_\beta(\boldsymbol{H}_m, \boldsymbol{x}_i)$ and the ground truth prediction $m_{\theta^*}(\boldsymbol{x}_i)$, where both these quantities are probabilities across different classes. The training problem is as follows:

$$\text{minimize}_\beta \ \sum_{i \in D, m \in \mathcal{M}} \text{KL}(m_{\theta^*}(\boldsymbol{x}_i) || g_\beta(\boldsymbol{H}_m, \boldsymbol{x}_i)) \quad (7)$$

**Generalization across architectures but not instances.** Note that the goal of the model approximator is to predict the output on $\boldsymbol{x}$ in the training set $D_{\text{tr}}$ for unseen architecture $m'$ so that using these predictions, our method can select the subset $S$ from $D_{\text{tr}}$ in a way that $m'$ trained on $S$ shows high accuracy on $D_{\text{test}}$. Since the underlying subset $S$ has to be chosen from the training set $D_{\text{tr}}$ for an arbitrary architecture $m'$, it is enough for the model approximator to mimic the model output only on the training set $D_{\text{tr}}$— it need not have to perform well in the test set $D_{\text{test}}$.

## 4.3 Subset sampler and design of transductive and inductive SUBSELNET

**Subset sampler.** We draw $S$, an ordered set of elements, using $\pi$ as follows. Having chosen the first $t$ instances $S_t = \{s_1, ..s_t\}$ from $D$ with $S_0 = \emptyset$, it draws the $(t+1)$-th element $(\boldsymbol{x}, y)$ from the remaining instances in $D$ with a probability proportional to $\exp(\pi(\boldsymbol{x}, y))$ and then repeat it for $b$ times. Thus, the probability of selecting the ordered set of elements $S = \{s_1, ..., s_b\}$ is given by

$$\Pr{}_\pi(S) = \prod_{t=0}^{b} \frac{\exp(\pi(\boldsymbol{x}_{s_{t+1}}, y_{s_{t+1}}))}{\sum_{s_\tau \in D \setminus S_t} \exp(\pi(\boldsymbol{x}_{s_\tau}, y_{s_\tau}))} \tag{8}$$

The optimization (3) suggests that once $\text{GNN}_\alpha$ and $g_\beta$ are trained, we can use them to approximate the output of the trained model $m_{\theta*}$ for an unseen architecture $m'$ and use it to compute $\pi$. Thus, this already removes a significant overhead of model training and facilitates fast computation of $\pi$, and further leads us to develop Transductive-SUBSELNET and Inductive-SUBSELNET based on how we can compute $\pi$, as described at the end of Section 3.1.

**Transductive-SUBSELNET.** The first variant of the model is transductive in terms of the computation of $\pi$. Once we train the architecture encoder and the model approximator, we compute $\pi$ by solving the optimization problem explicitly with respect to $\pi$ every time when we wish to select a data subset for a new architecture. Given trained model $\text{GNN}_{\hat{\alpha}}, g_{\hat{\beta}}$ and a new architecture $m' \in \mathcal{M}$, we solve the optimization problem to find the subset sampler $\Pr_\pi$ during inference time for a new architecture $m'$.

$$\min_\pi \mathbb{E}_{S \in \Pr_\pi(\bullet)} \Lambda(S; m'; \pi, g_{\hat{\beta}}, \text{GNN}_{\hat{\alpha}}) \tag{9}$$

Such an optimization still consumes time during inference. However, it is still significantly faster than the combinatorial methods [20, 19, 37, 47] thanks to sidestepping the explicit model training using a model approximator.

**Inductive-SUBSELNET.** In contrast to the transductive model, the inductive model does not require explicit optimization of $\pi$ in the face of a new architecture. To that aim, we approximate $\pi$ using a neural network $\pi_\psi$ which takes two signals as inputs— the dataset $D$ and the outputs of the model approximator for different instances $\{g_{\hat{\beta}}(\mathbf{H}_m, \boldsymbol{x}_i) \,|\, i \in D\}$ and finally outputs a score for each instance $\pi_\psi(\boldsymbol{x}_i, y_i)$. Here, the training of $\pi_\psi$ follows from the optimization (3):

$$\min_\psi \sum_{m \in \mathcal{M}} \mathbb{E}_{S \sim \Pr_{\pi_\psi}} \Lambda(S; m; \pi_\psi, g_{\hat{\beta}}, \text{GNN}_{\hat{\alpha}}) \tag{10}$$

Such an inductive model can select an optimal distribution of the subset that should be used to efficiently train any model $m_\theta$, without explicitly training $\theta$ or searching for the underlying subset.

**Architecture of $\pi_\psi$ for Inductive-SUBSELNET.** We approximate $\pi$ using $\pi_\psi$ using a neural network which takes three inputs – $(\boldsymbol{x}_j, y_j)$, the corresponding output of the model approximator, *i.e.*, $\boldsymbol{o}_{m, \boldsymbol{x}_j} = g_\beta(\text{GNN}_\alpha(G_m), \boldsymbol{x}_j)$ from Eq. (6) and the node representation matrix $\boldsymbol{H}_m$ and provides us a positive selection score $\pi_\psi(\boldsymbol{H}_m, \boldsymbol{x}_j, y_j, \boldsymbol{o}_{m, \boldsymbol{x}_j})$. In practice, $\pi_\psi$ is a three-layer feed-forward network containing Leaky-ReLU activation functions for the first two layers and sigmoid activation at the last layer.

## 4.4 Training and inference routines

**Training.** The training phase for both transductive and inductive variants, first utilizes the TRAINPIPELINE(Algorithm 1) routine to train the GNN (TRAINGNN), re-order the embeddings based on BFS ordering (BFS), train the model approximator (TRAINAPPROX), to obtain $\hat{\beta}$. TRAINTRANSDUCTIVE(Algorithm 2) routine doesn't require any further training, while the TRAININDUCTIVE(Algorithm 3) routine uses the TRAINPI to train $\psi$ for computing $\pi$.

**Inference.** Given a new architecture $m'$, our goal is to select a subset $S$, with $|S| = b$ which would facilitate efficient training of $m'$. Given trained SUBSELNET, we compute $\boldsymbol{H}_{m'} = \text{GNN}_{\hat{\alpha}}(G_{m'})$, compute the model approximator output $g_{\hat{\beta}}(\boldsymbol{H}_{m'})$. Using them we compute $\pi$ for

---

**Algorithm 1** Training Pipeline

1: **function** TRAINPIPELINE($D, \mathcal{M}, \{\theta^*\}$)
2:    $\hat{\alpha} \leftarrow$ TRAINGNN($\mathcal{M}$)
3:    **for** $m \in \mathcal{M}_{\text{tr}}$ **do**
4:       $\boldsymbol{H}_m \leftarrow \text{GNN}_{\hat{\alpha}}(m)$, pos $\leftarrow$ BFS($G_m, \boldsymbol{H}_m$)
5:    $\hat{\beta} \leftarrow$ TRAINAPPROX($\boldsymbol{H}_m, \{\boldsymbol{x}_i\}, \text{pos}, \{\theta^*\}$)
6:    **return** $\hat{\alpha}, \hat{\beta}, \boldsymbol{H}_m$

---

**Algorithm 2** Transductive Procedure

1: **function** TRAINTRANSDUCTIVE($D, \mathcal{M}, \{\theta^*\}$)
2:    $\hat{\alpha}, \hat{\beta}, \boldsymbol{H}_m \leftarrow$ TRAINPIPELINE($D, \mathcal{M}, \{\theta^*\}$)

1: **function** INFERTRANSDUCTIVE($D, \hat{\alpha}, \hat{\beta}, m'$)
2:    $\pi^* \leftarrow \min_\pi \mathbb{E}_{S \in \Pr_\pi(\bullet)} \Lambda(S; m'; \pi, g_{\hat{\beta}}, \text{GNN}_{\hat{\alpha}})$
3:    $S^* \sim \Pr_{\pi^*}(\bullet)$
4:    TRAINNEWMODEL($m'; S^*$)

---

Transductive-SUBSELNET by explicitly solving the optimization problem stated in Eq. 9 and draw $S \sim \mathrm{Pr}_\pi(\bullet)$. For the inductive variant, we draw $S \sim \mathrm{Pr}_{\pi_{\hat{\psi}}}(\bullet)$ where $\hat{\psi}$ is the learned value of $\psi$.

Given an unseen architecture $m'$ and trained parameters of SUBSELNET, *i.e.*, $\hat{\alpha}, \hat{\beta}$ and $\hat{\psi}$, the INFERTRANSDUCTIVE(Algorithm 2) routine solves the optimization problem on $\pi$ explicitly to compute $\pi$, where $\Lambda(\cdot)$ is defined in Eq. (2).

---

**Algorithm 3** Inductive Procedure

1: **function** TRAININDUCTIVE($D, \mathcal{M}, \{\theta^*\}$)
2: $\quad \hat{\alpha}, \hat{\beta}, \boldsymbol{H}_m \leftarrow$ TRAINPIPELINE($D, \mathcal{M}, \{\theta^*\}$)
3: $\quad \boldsymbol{o} \leftarrow [g_{\hat{\beta}}(\{\boldsymbol{H}_m, \boldsymbol{x}_i\})]_{i,m}$
4: $\quad \hat{\psi} \leftarrow$ TRAINPI($\boldsymbol{o}, \{\boldsymbol{H}_m\}, \{\boldsymbol{x}_i\}$)

1: **function** INFERINDUCTIVE($D, \hat{\alpha}, \hat{\beta}, \hat{\psi}, m'$)
2: $\quad$ Compute $\pi_{\hat{\psi}}(\boldsymbol{H}_{m'}, \boldsymbol{x}_i, y_i, \boldsymbol{o}_{m', \boldsymbol{x}_i}) \; \forall i \in D$
3: $\quad S^* \sim \mathrm{Pr}_{\pi_{\hat{\psi}}}(\bullet)$
4: $\quad$ TRAINNEWMODEL($m'; S^*$)

---

INFERINDUCTIVE (Algorithm 3) utilizes $\hat{\psi}$, *i.e.*, trained parameters from the subset sampler to compute $\pi_{\hat{\psi}}$. Then the subset $S^*$ is drawn from $\pi$ or $\pi_\psi$ and is used to train $m'$ using TRAINNEWMODEL.

# 5 Experiments

In this section, we provide comprehensive evaluation of SUBSELNET against several strong baselines on five real world datasets. In Appendix E, we present additional results. Our code is in `https://github.com/structlearning/subselnet`.

## 5.1 Experimental setup

**Datasets.** We use FMNIST [56], CIFAR10 [26], CIFAR100 [25], Tiny-Imagenet-200 [27] and Caltech-256 [13] (Cal-256). Cal-256 has imbalanced class distribution; the rest are balanced. We transform an input image $\boldsymbol{X}_i$ to a vector $\boldsymbol{x}_i$ of dimension 2048 by feeding it to a pre-trained ResNet50 v1.5 model [16] and use the output from the penultimate layer as the image representation.

**Model architectures and baselines.** We use model architectures from NAS-Bench-101 [61] in our experiments. We compare Transductive-SUBSELNET and Inductive-SUBSELNET against three non-adaptive subset selection methods – (i) Facility location [11, 17] where we maximize $FL(S) = \sum_{j \in D} \max_{i \in S} \boldsymbol{x}_i^\top \boldsymbol{x}_j$ to find $S$, (ii) Pruning [48], and (iii) Selection-via-Proxy [5] and four adaptive subset selection methods – (iii) Glister [20], (iv) Grad-Match [19], (v) EL2N [42] and (vi) GraNd [42]. The non-adaptive subset selectors select the subset before the training begins and thus, never access the rest of the training set again during the training iterations. On the other hand, the adaptive subset selectors refine the choice of subset during training iterations and thus they need to access the full training set at each training iteration. Appendix D contains additional details about the baselines and Appendix E contains experiments with more baselines.

**Evaluation protocol.** We split the model architectures $\mathcal{M}$ into 70% training ($\mathcal{M}_{\mathrm{tr}}$), 10% validation ($\mathcal{M}_{\mathrm{val}}$) and 20% test ($\mathcal{M}_{\mathrm{test}}$) folds. However, training model approximator requires supervision from the pre-trained models $m_{\theta^*}$. Pre-training large number of models can be expensive. Therefore, we limit the number of pre-trained models to a diverse set of size 250, that ensures efficient representation over low-parameter and high-parameter regimes, and using more than this showed no visible advantage. We show the parameter statistics in Appendix D. However, for the architecture encoder, we use the entire set $\mathcal{M}_{\mathrm{tr}}$ for GNN training. We split the dataset $D$ into $D_{\mathrm{tr}}, D_{\mathrm{val}}$ and $D_{\mathrm{test}}$ in the similar 70:10:20 folds. We present $\mathcal{M}_{\mathrm{tr}}, \mathcal{M}_{\mathrm{val}}, D_{\mathrm{tr}}$ and $D_{\mathrm{val}}$ to our method and estimate $\hat{\alpha}, \hat{\beta}$ and $\hat{\psi}$ (for Inductive-SUBSELNET model). None of the baseline methods supports any generalizable learning protocol for different architectures and thus cannot leverage the training architectures during test. Given an architecture $m' \in \mathcal{M}_{\mathrm{test}}$, we select the subset $S$ from $D_{\mathrm{tr}}$ using our subset sampler ($\mathrm{Pr}_\pi$ for Transductive-SUBSELNET or $\mathrm{Pr}_{\pi_{\hat{\psi}}}$ for Inductive-SUBSELNET). Similarly, all the non-adaptive subset selectors select $S \subset D_{\mathrm{tr}}$ using their own algorithms. Once $S$ is selected, we train the test models $m' \in \mathcal{M}_{\mathrm{test}}$ on $S$. We perform our experiments with different $|S| = b \in (0.005|D|, 0.9|D|)$ and compare the performance between different methods using three quantities: (1) Relative Accuracy Reduction (RAR) computed as the drop in test accuracy on training with a chosen subset as compared to training with the entire dataset, i.e, $\mathrm{RAR}(S, D) = \frac{1}{|\mathcal{M}_{\mathrm{test}}|} \sum_{m' \in \mathcal{M}_{\mathrm{test}}} (1 - \mathrm{Acc}(m' \,|\, S)/\mathrm{Acc}(m' \,|\, D))$ where $\mathrm{Acc}(m' \,|\, X)$ denotes the test accuracy when $m'$ is trained on the set $X$. Lower RAR indicates better performance. (2) Computational efficiency, *i.e.*, the speedup achieved with respect to training with full dataset. It is measured with respect to $T_f / T$. Here, $T_f$ is the time taken for training with full dataset; and, $T$ is the time taken for the entire inference task, which is the average time for selecting

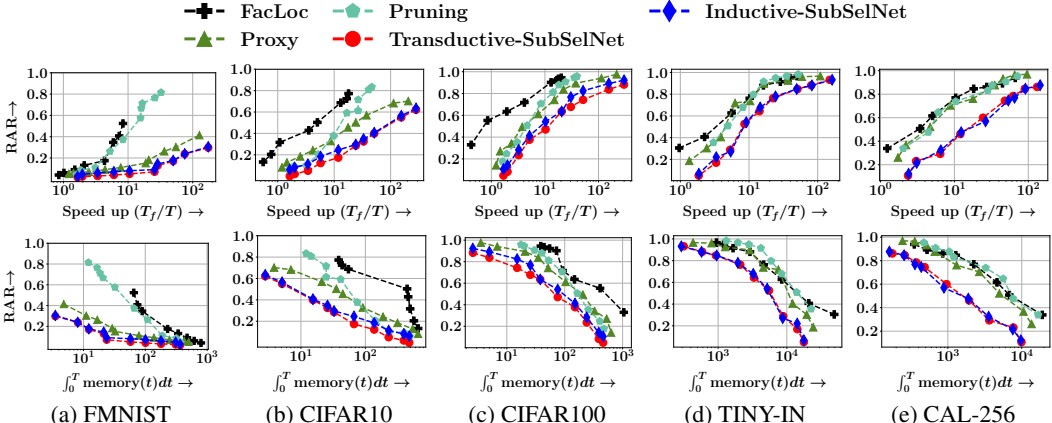

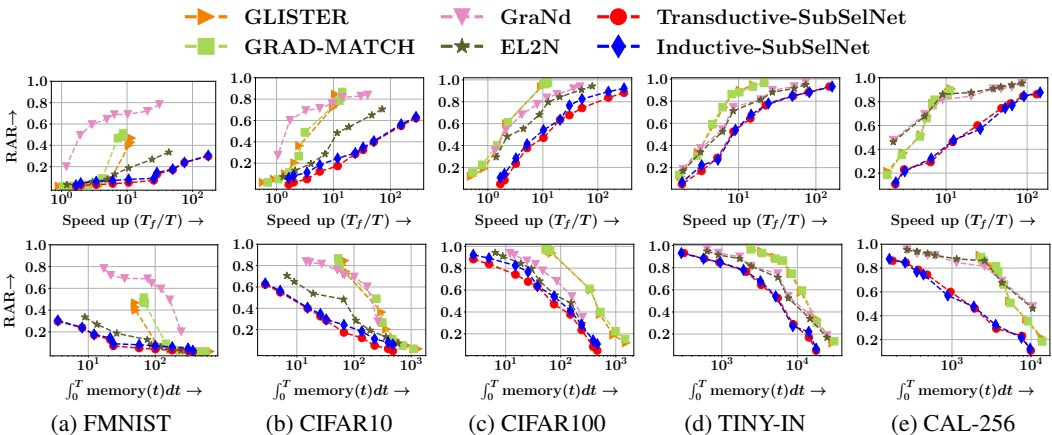

subsets across the test models $m' \in \mathcal{M}_{\text{test}}$ plus the average training time of these test models on the respective selected subsets. (3) Resource efficiency in terms of the amount of memory consumed during the entire inference task, described in item (2), which is measured as $\int_0^T \text{memory}(t)\, dt$ where memory($t$) is amount of memory consumed at timestamp $t$ in the unit of GB-min.

## 5.2 Results

**Comparison with baselines.** Here, we compare different methods in terms of the trade-off between Relative accuracy reduction RAR (lower is better) and computational efficiency as well as RAR and resource efficiency. In Figures 2 and 3, we probe the variation between these quantities by varying the size of the selected subset $|S| = b \in (0.005|D|, 0.9|D|)$ for non-adaptive and adaptive baselines, respectively. We make the following observations. **(1)** Our methods trade-off between accuracy vs. computational efficiency as well as accuracy vs. resource efficiency more effectively than all the methods, including the adaptive methods which refine their choice of subset as the model training progresses. **(2)** In FMNIST, our method achieves 10% RAR at ∼4.4 times the speed-up and using 77% lesser memory than EL2N, the best baseline (Table 1, tables for other datasets are in Appendix E). **(3)** There is no consistent winner across baselines. However, Glister and Grad-Match mostly remain among top three baselines, across different methods. In particular, they outperform others in Tiny-Imagenet and Cal-256, in high accuracy (low RAR) regime.

| | Speedup | | Memory | |
|---|---|---|---|---|
| **RAR** | 10% | 20% | 10% | 20% |
| GLISTER | 5.64 | 7.85 | 116.36 | 98.51 |
| GradMatch | 4.17 | 5.24 | 243.75 | 136.40 |
| EL2N | 6.50 | 16.42 | 139.89 | 77.63 |
| Inductive | 28.64 | 69.24 | 22.73 | 8.24 |
| Transductive | 28.63 | 68.36 | 21.25 | 8.24 |

Table 1: Speedup and memory (GB-min) in reaching 10% and 20% RAR on FMNIST

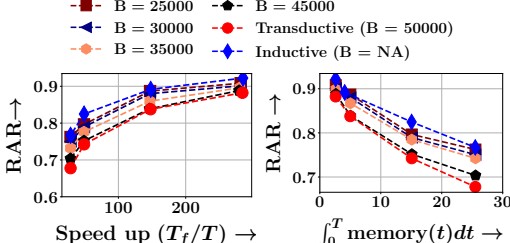

Figure 4: Hybrid-SUBSELNET

**Hybrid-SUBSELNET.** In FMINST, CIFAR10 and CIFAR100, we observe that Transductive-SUBSELNET offers better traded off than Inductive-SUBSELNET. Here, we design a hybrid version of our model, called as Hybrid-SUBSELNET and evaluate it on a regime where the gap between transductive and inductive SUBSELNET is significant. One of such regimes is the part of the trade-off plot in CIFAR100, where the speed up $T_f/T \geq 28.09$ (Figures 2 and 3). Here, given the budget of the subset $b$, we first choose $B > b$ instances using Inductive-SUBSELNET and the final $b$ instances by running the explicit optimization routines in Transductive-SUBSELNET. Figure 4 shows the results for $B = \{25K, 30K, 35K, 45K, 50K\}$. We observe that Hybrid-SUBSELNET allow us to smoothly trade off between Inductive-SUBSELNET and Transductive-SUBSELNET, by tuning $B$. It allows us to effectively use resource-constrained setup with limited GPU memory, wherein the larger subset $B$ can be selected using Inductive-SUBSELNET on a CPU, and the smaller *refined* subset $b$ can then be selected by solving transductive variant on GPU.

**Ablation study.** Here, we experiment with three candidates of model approximator $g_\beta$ ( Feedforward, LSTM and our proposed attention based approximator) with three different subset samplers $\pi$ (uncertainty based, loss based and our proposed subset sampler). Thus, we have nine different combinations of model approximator and subset selection strategies. In the uncertainty and loss based subset samplers, we take top-$b$ instances based on the uncertainty and loss. We measure uncertainty using the entropy of the predicted distribution of the target classes. We compare the performance in terms of the test RAR of the test architectures. Moreover, we also evaluate the model approximator $g_\beta$ alone — without the presence of the subset sampler — using KL divergence between the gold model outputs and predicted model outputs on the training instances $\frac{1}{|D_{tr}||\mathcal{M}_{test}|} \sum_{i \in D_{tr}, m \in \mathcal{M}_{test}} \mathrm{KL}(m_{\theta^*}(\boldsymbol{x}_i) || g_\beta(\boldsymbol{H}_m, \boldsymbol{x}_i))$. Table 3 summarizes the results for 3%, 5% and 10% subsets for CIFAR10. We make the following observations: (1) The complete design of our method, i.e., Our model approximator (Transformer) + Our subset sampler (SUBSELNET) performs best in terms of RAR. (2) Our neural-network for model approximator mimics the trained model output better than LSTM and Feedforward architectures.

*Can model approximator substitute our subset selector pipeline?* The task of the model approximator $g_\beta$ is to predict accuracy for unseen architecture. Then, a natural question is that is it possible to use the model approximator to directly predict accuracy of the unseen architecture $m'$, instead of using such long pipeline to select subset $S$ followed with training on $S$. However, as discussed in the end of Section 4.2, the model approximator $g_\beta$ is required to generalize across unseen architectures but not the unseen instances, as its task is to help select the *training* subset. Table 3 already showed that $g_\beta$ closely mimics the output of the trained model for the unseen architecture $m' \in \mathcal{M}_{test}$ and on training instances $\boldsymbol{x}$ (KL div. column). Here, we investigate the performance of $g_\beta$ on the test instances and test architectures.

| $b$ (in % ) | 90% | 70% | 20% |
|---|---|---|---|
| RAR(our $\mid S$) - RAR($g_\beta$) | -0.487 | -0.447 | -0.327 |

Table 2: RAR using $g_\beta$ on CIFAR10

Table 2 shows that the performance of $g_\beta$ on the test instances is significantly poorer than our method. This is intuitive as generalizing both the model space and the instance space is extremely challenging, and we also do not need it in general.

**Using SUBSELNET in AutoML.** AutoML-related tasks can be significantly sped-up when we replace the entire dataset with a representative subset. Here, we apply SUBSELNET to two AutoML applications: Neural Architecture Search (NAS) and Hyperparameter Optimization (HPO).
*Neural Architecture Search:* We apply our method on DARTS architecture space to search for an architecture using subsets. During this search process, at each iteration, the underlying network is traditionally trained on the entire dataset. In contrast, we train this underlying network on the subset returned by our method for this architecture. Following Na et al. [39], we report test misclassification

| Design choice of $g_\beta$ and $\pi$ | RAR | | | KL-div |
|---|---|---|---|---|
| | $b = 0.03\|D\|$ | $b = 0.05\|D\|$ | $b = 0.1\|D\|$ | (does not depend on $b$) |
| Feedforward ($g_\beta$)+ Uncertainty ($\pi$) | 0.657 | 0.655 | 0.547 | |
| Feedforward ($g_\beta$)+ Loss ($\pi$) | 0.692 | 0.577 | 0.523 | 0.171 |
| Feedforward + Inductive (our) ($\pi$) | 0.451 | 0.434 | 0.397 | |
| LSTM ($g_\beta$)+ Uncertainty ($\pi$) | 0.566 | 0.465 | 0.438 | |
| LSTM ($g_\beta$)+ Loss ($\pi$) | 0.705 | 0.541 | 0.455 | 0.102 |
| LSTM ($g_\beta$)+ Inductive (our) ($\pi$) | 0.452 | 0.412 | 0.386 | |
| Attn. (our) ($g_\beta$)+ Uncertainty ($\pi$) | 0.794 | 0.746 | 0.679 | |
| Attn. (our) ($g_\beta$)+ Loss ($\pi$) | 0.781 | 0.527 | 0.407 | **0.089** |
| Attn. (our) ($g_\beta$)+ Inductive (our) ($\pi$) | **0.429** | **0.310** | **0.260** | |

Table 3: RAR and KL-divergence for different $g_\beta + \pi$ on CIFAR10 for 3%, 5% and 10% subset sizes

| $b$ (in %) | 10% | 20% | 40% |
|---|---|---|---|
| Full | | 2.78 | |
| Random | 3.02 | 2.88 | 2.96 |
| Proxy [39] | 2.92 | 2.87 | 2.88 |
| Our | **2.82** | **2.76** | **2.68** |

Table 4: Test Error (%) on architecture given by NAS on CIFAR10

| Method | $b = 5\%$ | | $b = 10\%$ | |
|---|---|---|---|---|
| | TE | S/U | TE | S/U |
| Full | 2.48 | 1 | 2.48 | 1 |
| Random | 5.4 | **16.66** | 3.72 | **11.29** |
| AUTOMATA | 5.26 | 0.51 | 3.39 | 0.20 |
| Our | **4.11** | 16.11 | **2.70** | 10.96 |

Table 5: Test Error (%) (TE) and Speed-up (S/U) for the hyperparameters selected by HPO on CIFAR10

| Method | # Test Architectures | | |
|---|---|---|---|
| | 200 | 300 | 400 |
| Full | 7111 | 7111 | 7111 |
| GLISTER | 3419 | 3419 | 3419 |
| GRAD-MATCH | 2909 | 2909 | 2909 |
| Our | **1844** | **1635** | **1496** |

Table 6: Amortization cost (seconds) after querying test architectures on CIFAR10

error of the architecture which is selected by the corresponding subset selector guided NAS methods, *i.e.*, our method (transductive), random subset selection (averaged over 5 runs) and proxy-data [39]. Table 4 shows that our method performs better than the baselines.

*Hyperparameter Optimization:* Finding the best set of hyperparameters from their search space for a model is computationally intensive. We look at speeding-up the tuning process by searching the hyperparameters while training the model on a small representative subset $S$ instead of $D$. Following Killamsetty et al. [22], we consider optimizer and scheduler specific hyperparameters and report average test misclassification error across the models trained on optimal hyperparameter choice returned by our method (transductive), random subset selection (averaged over 5 runs) and AUTOMATA [22]. Table 5 shows that we are outperforming the baselines in terms of accuracy-speedup tradeoff. Appendix D contains more details about the implementation.

**Amortization Analysis.** Figures 2 and 3 show that our method is substantially faster than the baselines during inference, once we have our neural pipeline trained. Such inference time speedup is the focus of many other applications, E.g., complex models like LLMs are difficult and computationally intensive, but their inference is fast for several queries once trained. However, we recognize that there is a computational overhead in training our model, arising due to the pre-training of the models $m_{\theta*}$ used for supervision. Since the prior training is only a one-time overhead, the overall cost is amortized by querying multiple architectures for their subsets. We measure amortized cost $T_{\text{total}}/M_{\text{total}}$ (time in seconds), where $T_{\text{total}}$ is the total time used from beginning of the pipeline to end of reporting final accuracy on the test architectures and $M_{\text{total}}$ is the total number of training and test architectures. Table 6, shows the results for 10% subset on the top baselines for CIFAR10, which shows that the training overhead of our method (transductive) quickly diminishes with number of test architectures.

# 6 Conclusion

In this work, we develop SUBSELNET, a subset selection framework, which can be trained on a set of model architectures, to be able to predict a suitable training subset before training a model, for an unseen architecture. To do so, we first design a neural architecture encoder and model approximator, which predicts the output of a new candidate architecture without explicitly training it. We use that output to design transductive and inductive variants of our model.

**Limitations.** The SUBSELNET pipeline offers quick inference-time subset selection but a key limitation of our method is that it entails a pre-training overhead, although its overhead vanishes as we query more architectures. Such expensive training can be reduced by efficient training methods [65]. In future, it would be interesting to incorporate signals from different epochs with a sequence encoder to train a subset selector. Apart from this, our work does not assume the distribution shift of architectures from training to test. If the architectures vary significantly from training to test, then there is significant room for performance improvement.

**Acknowledgements.** Abir De acknowledges the Google Research gift funding.

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

# Efficient Data Subset Selection to Generalize Training Across Models: Transductive and Inductive Networks (Appendix)

## A    Limitations

While our work outperforms several existing subset selection methods, it suffers from three key limitations.

(1) We acknowledge that there indeed is a computation time for pre-training the model approximator. However, as we mentioned in amortization analysis in Section 5.2, the one-time overhead is offset quickly by the speed and effectiveness of the following selection pipeline for the subset selection of unseen architectures. Since the prior training is only a one-time overhead, the overall cost is amortized by the number of unseen architectures during inference time training. In practice, when it is used to predict the subset for a large number of unseen architectures, the effect of training overhead quickly vanishes. As demonstrated by our experiments (Figures 2 and 3, Table 6), once the pipeline of all the neural networks is set up, the selection procedure is remarkably fast and can be easily adapted for use with unseen architectures.

To give an analogy, premier search engines invest a lot of resources in making fast inferences rather than training. They build complex models that are difficult and computationally intensive, but their inference is fast for several queries once trained. Thus, the cost is amortized by the large number of queries. Another example is locality-sensitive hashing. Researchers design trainable models for LSH whose purpose is to make fast predictions. Training LSH models can take a lot of time, but again this cost is amortized by the number of unseen queries.

Finally, we would like to highlight many efficient model training methods without complete training (running a few epochs via curriculum learning [66]), which one can easily explore and plug with our method for a larger dataset like Imagenet-1K.

(2) We use the space of neural architectures which comprises only of CNNs. We did not experiment with sequence models such as RNNs or transformers. However, we believe that our work can be extended with RNNs or transformer based architectures.

(3) If the distribution of network architectures varies widely from training to test, then there is significant room for performance improvement. In this context, one can develop domain adaptation methods for graphs to tackle different out-of-distribution architectures more effectively.

## B    Broader Impact

Our work can be used to provide significant compute efficiency by the trainable subset selection method we propose. It can be used to save a lot of time and power, that ML model often demands. Specifically, it can be used in the following applications in the context of AutoML.

**Fast tuning of hyperparameters related to optimizer/training.** Consider the case where we need to tune non-network hyperparameters, such as learning rate, momentum, and weight decay. Given the architecture, we can choose the subset obtained using our method to train the underlying model parameters for different hyperparameters, which can then be used for cross-validation. Note that we would use the same subset in this problem since the underlying model architecture is fixed, and we obtain a subset for the given architecture independent of the underlying non-network hyperparameters. We have shown utility of our method in our experiments in Section 5.2.

**Fast tuning of model related hyperparameters.** Consider the case where we need to tune network-related hyperparameters, such as the number of layers, activation functions, and the width of intermediate layers. Instead of training each instance of these models on the entire data, we can train them on the subset of data obtained from our method to quickly obtain the trained model, which can then be used for cross-validation.

**Network architecture search.** As we shown in our experiments, our method can provide speedup in network architecture search. Here, instead of training the network the entire network during

architecture exploration, we can restrict the training on a subset of data, which can provide significant speedup.

Note that, the key goal of our method is design a trainable subset selection method that generalizes across architectures. As we observed in our experiments, these methods can be useful in the above applications— however, our method is a generic framework and not tailored to any one of the above applications. Therefore, our method may need application specific modifications before directly deploying it practice. However, our method can serve as a base model for the practitioner who intends to speed up for one of the above applications.

We do not foresee any negative social impact of our work.

## C   Additional discussion on related work

Our work is closely related to representation learning for model architectures, network architecture search, data subset selection.

**Representation learning for model architectures.** Recent work in network representation learning use GNN based encoder-decoder to encapsulate the local structural information of a neural network into a fixed-length latent space [64, 40, 59, 34]. By employing an asynchronous message passing scheme over the directed acyclic graph (DAG), GNN-based methods model the propagation of input data over the actual network structure. Apart from encodings based solely on the structure of the network, White et al. [55], Yan et al. [60] produce computation-aware encodings that map architectures with similar performance to the same region in the latent space. Following the work of Yan et al. [59], we use a graph isomorphism network as an encoder but instead of producing a single graph embedding, our method produces a collection of node embeddings, ordered by breadth-first-search (BFS) ordering of the nodes. Our work also differs in that we do not employ network embeddings to perform downstream search strategies. Instead, architecture embeddings are used in training a novel *model approximator* that predicts the logits of a particular architecture, given an architecture embedding and a data embedding.

**Machine learning on architecture space.** We use NAS-Bench-101 in our method. This dataset was built in the context of network architecture search (NAS). The networks discovered by NAS methods often come from an underlying search space, usually designed to constrain the search space size. One such method is to use cell-based search spaces [35, 68, 30, 43, 61, 9]. Although we utilize the NAS-Bench-101 search space for architecture retrieval, our work is fundamentally different from NAS. In contrast to the NAS methods, which search for the best possible architecture from the search space using either sampling or gradient-descent based methods [1, 67, 44, 45, 31, 49], our work focuses on efficient data subset selection given a dataset and an architecture, which is sampled from a search space. Our work utilizes graph representation learning on the architectures sampled from the mentioned search spaces to project an architecture under consideration to a continuous latent space, utilize the model expression from the latent space as proxies for the actual model and proceed with data subset selection using the generated embedding, model proxy and given dataset.

**Data subset selection.** Data subset selection is widely used in literature for efficient learning, coreset selection, human centric learning, *etc*. Several works cast the efficient data subset selection task as instance of submodular or approximate-submodular optimization problem [19, 51–53, 20, 47]. Another line of work focus on selecting coresets which are expressed as the weighted combination of subset of data, approximating some characteristics, *e.g.*, loss function, model prediction [10, 37, 15, 4, 33]. Among other works, Toneva et al. [50] showed coresets can be selected by omitted several instances based on the forgetting dynamics at the time of training. Na et al. [39] selects proxy data based on entropy of the model. Coleman et al. [5] uses proxy model and then use it for coreset selection. Guo et al. [14] develop a library on coreset selection.

Our work is closely connected to simultaneous model learning and subset selection [7, 6, 47]. These existing works focus on jointly optimizing the training loss, with respect to the subset of instances and the parameters of the underlying model. Among them [7, 6] focus on distributing decisions between human and machines, whereas [47] aims for efficient learning. However, these methods adopt a combinatorial approach for selecting subsets and consequently, they are not generalizable across architectures. In contrast, our work focuses on differentiable subset selection mechanism, which can generalize across architectures.

# D  Additional details about experimental setup

## D.1  Dataset

**Datasets** $(D)$**.**

| Dataset | No. of Classes | Imbalanced | Train-Test Split | Shape | Transformations Applied |
|---|---|---|---|---|---|
| FMNIST | 10 | ✗ | (60K,10K) | 28x28x1 | Normalize |
| CIFAR10 | 10 | ✗ | (50K,10K) | 32x32x3 | RandomHorizontalFlip, RandomCrop, Normalize |
| CIFAR100 | 100 | ✗ | (50K,10K) | 32x32x3 | RandomHorizontalFlip, RandomCrop, Normalize |
| Tiny-Imagenet | 200 | ✗ | (100K,10K) | 64x64x3 | RandomHorizontalFlip, RandomVerticalFlip, Normalize |
| Caltech-256 | 257 | ✓ | (24.5K,6.1K) | 96x96x3 | RandomHorizontalFlip, Resize, Normalize |

Table 7: A brief description of the datasets used along with the transformations applied during training

**Architectures** $(\mathcal{M})$**.** We leverage the NASBench-101 search space as an architecture pool. It consists of $423,624$ unique architectures with the following constraints – (1) number of nodes in each cell is at most 7, (2) number of edges in each cell is at most 9, (3) barring the input and output, there are three unique operations, namely $1 \times 1$ convolution, $3 \times 3$ convolution and $3 \times 3$ max-pool. We utilize the architectures from the search space in generating the sequence of embeddings along with sampling architectures for the training and testing of the encoder and datasets for the subset selector. As mentioned in the experimental setup, pre-training large number of models can be expensive.

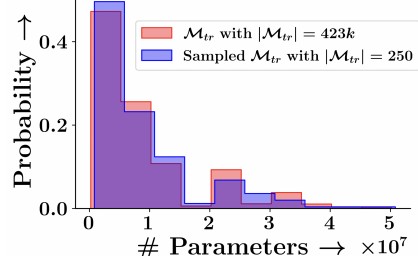

Figure 5: Distribution of parameters of architectures in $\mathcal{M}_{\text{tr}}$ when $|\mathcal{M}_{\text{tr}}| = 423k$ (blue), and $\mathcal{M}_{\text{tr}}$ with the sampled set of 250 architectures (orange).

Therefore, we choose a diverse subset of architectures from $\mathcal{M}_{\text{tr}}$ of size 250 that ensures efficient representation over low-parameter and high-parameter regimes. The distributions of true and sampled architectures are given in Figure 5.

Note that the pre-training can be made faster by efficient model training methods without complete training (running a few epochs via curriculum learning [66]), which can be easily plugged with our method.

## D.2  Implementation details about baselines

**Facility Location (FL).** We implemented facility location on all the three datasets using the apricot [1] library. The similarity matrix was computed using Euclidean distance between data points, and the objective function was maximized using the naive greedy algorithm.

**Pruning.** It selects a subset from the entire dataset based on the uncertainty of the datapoints while partial training. In our setup, we considered ResNet-18 as a master model, which is trained on each dataset for 5 epochs. Post training, the uncertainty measure is calculated based on the probabilities of each class, and the points with highest uncertainty are considered in the subset. We train the master model at a learning rate of 0.025.

**Proxy.** It selects a subset

**Glister and Grad-Match.** We implemented GLISTER [20] and Grad-Match [19] using the CORDS library. We trained the models for 50 epochs, using batch size of 20, and selected the subset after every 10 epochs. The loss was minimized using SGD with learning rate of 0.01, momentum of 0.9 and weight decay with regularization constant of $5 \times 10^{-4}$. We used cosine annealing for scheduling

---

[1] https://github.com/jmschrei/apricot

the learning rate with $T_{max}$ of 50 epochs, and used $10\%$ of the training data as the validation set. Details of specific hyperparameters for stated as follows.

*Glister* uses a greedy selection approach to minimize a bi-level objective function. In our implementation, we used stochastic greedy optimization with learning rate $0.01$, applied on the data points of each mini-batch. Online-Glister approximates the objective function with a Taylor series expansion up to an arbitrary number of terms to speed up the process; we used 15 terms in our experiments.

*Grad-Match* applies the orthogonal matching (OMP) pursuit algorithm to the data points of each mini-batch to match gradient of a subset to the entire training/validation set. Here, we set the learning rate is set to $0.01$. The regularization constant in OMP is $1.0$ and the algorithm optimizes the objective function within an error margin of $10^{-4}$.

**GraNd.** This is an adaptive subset selection strategy in which the norm of the gradient of the loss function is used as a score to rank a data point. The gradient scores are computed after the model has trained on the full dataset for the first few epochs. For the rest of epochs, the model is trained only on the top-$k$ data points, selected using the gradient scores. In our implementation, we let the model train on the full dataset for the first 5 epochs, and computed the gradient of the loss only with respect to the last layer fully connected layer.

**EL2N.** When the loss function used to compute the GraNd scores is the cross entropy loss, the norm of the gradient for a data point $\mathbf{x}$ can be approximated by $\mathbb{E}||p(\mathbf{x}) - y||_2$, where $p(\mathbf{x})$ is the discrete probability distribution over the classes, computed by taking `softmax` of the logits, and $y$ is the one-hot encoded true label corresponding to the data point $\mathbf{x}$. Similar to our implementation of GraNd, we computed the EL2N scores after letting the models train on the full data for the first 5 epochs.

### D.3 Implementation details about our model

**GNN$_\alpha$.** As we utilize NASBench-101 space as the underlying set of neural architectures, each computational node in the architecture can comprise of one of five *operations* and the one-hot-encoded feature vector $\mathbf{f}_u$. Since the set is cell-based, there is an injective mapping between the neural architecture and the cell structure. We aim to produce a sequence of embeddings for the cell, which in turn corresponds to that of the architecture. For each architecture, we use the initial feature $\mathbf{f}_u \in \mathbb{R}^5$ in as a five dimensional one-hot encoding for each operation. This is fed into INITNODE to obtain an 16 dimensional output. Here, INITNODE consists of a $5 \times 16$ linear, ReLU and $16 \times 16$ linear layers cascaded with each other. Each of EDGEEMBED and UPDATE consists of a $5 \times 128$ linear-BatchNorm-ReLU cascaded with a $128 \times 16$ linear layer. Moreover, the symmetric aggregator is a sum aggregator.

We repeat this layer $K$ times, and each iteration gathers information from $k < K$ hops. After all the iterations, we generate an embedding for each node, and following [62] we use the BFS-tree based node-ordering scheme to generate the sequence of embeddings for each network.

The GVAE-based architecture was trained for 10 epochs with the number of recursive layers $K$ set to 5, and the Adam optimizer was used with learning rate of $10^{-3}$. The entire search space was considered as the dataset, and a batch-size of 32 was used. Post training, we call the node embeddings collectively as the architecture representation.

To train the latent space embeddings, the parameters $\alpha$ are trained in an encoder-decoder fashion using a variational autoencoder. The mean $\mu$ and variance $\sigma$ on the final node embeddings $\boldsymbol{h}_u$ are:

$$\mu = \text{FCN}\left(\left[\boldsymbol{h}_u\right]_{u \in V_m}\right) \text{ and } \sigma = \exp\left(\text{FCN}\left(\left[\boldsymbol{h}_u\right]_{u \in V_m}\right)\right)$$

The decoder aims to reconstruct the original cell structure (i.e the nodes and the corresponding operations), which are one-hot encoded. It is modeled using single-layer fully connected networks followed by a sigmoid layer.

**Model Approximator $g_\beta$.** The model approximator $g_\beta$ is essentially a single-head attention block that acts on a sequence of node embeddings $\boldsymbol{H}_m = \{h_u | u \in V_m\}$. The Query, Key and Value are three linear networks with parameters: $\boldsymbol{W}_{\text{query}}$, $\boldsymbol{W}_{\text{key}}$ and $\boldsymbol{W}_{\text{value}} \in \mathbb{R}^{16 \times 8}$. Note that the matrix $\boldsymbol{W}_C \in \mathbb{R}^{8 \times 16}$ in Eq. (5). As described in Section 4.2, for each node $u$, we use a feedforward network, preceded and succeeded by layer normalization operations, which are given by the following set of

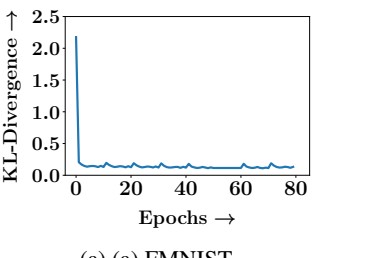
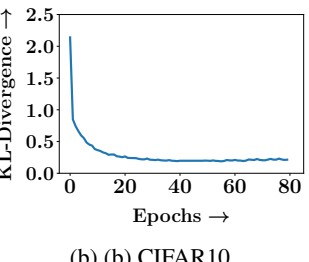

(a) (a) FMNIST

(b) (b) CIFAR10

Figure 6: Kullback-Leibler divergence values $(\text{KL}(m_{\theta^*}(\boldsymbol{x}_i) \,\|\, g_\beta(\boldsymbol{H}_m, \boldsymbol{x}_i)))$ computed during the training of the model encoder $g_\beta$ over 80 epochs.

equations (where LN the denotes Layer-Normalization operation):

$$\boldsymbol{\zeta}_{u,1} = \text{LN}(\text{Att}_u + \boldsymbol{h}_u; \gamma_1, \gamma_2),$$
$$\boldsymbol{\zeta}_{u,2} = \boldsymbol{W}_2^\top \text{RELU}(\boldsymbol{W}_1^\top \boldsymbol{\zeta}_{u,1}),$$
$$\boldsymbol{\zeta}_{u,3} = \text{LN}(\boldsymbol{\zeta}_{u,1} + \boldsymbol{\zeta}_{u,2}; \gamma_3, \gamma_4)$$

The fully connected network acting on $\zeta_{u,1}$ consists of matrices $W_1 \in \mathbb{R}^{16 \times 64}$ and $W_2 \in \mathbb{R}^{64 \times 16}$. All the trainable matrices along with the layer normalizations were implemented using the `Linear` and `LayerNorm` functions in Pytorch. The last item of the output sequence $\zeta_{u,3}$ is concatenated with the data embedding $\boldsymbol{x}_i$ and fed to another 2-layer fully-connected network with hidden dimension 256 and dropout probability of 0.3. The model approximator is trained by minimizing the KL-divergence between $g_\beta(\boldsymbol{H}_m, \boldsymbol{x}_i)$ and $m_{\theta^*}(\boldsymbol{x}_i)$. We used an AdamW optimizer with learning rate of $10^{-3}$, $\epsilon = 10^{-8}$, `betas` $= (0.9, 0.999)$ and weight decay of 0.005. We also used Cosine Annealing to decay the learning rate, and used gradient clipping with maximum norm set to 5. Figure 6 shows the convergence of the outputs of the model approximator $g_\beta(\boldsymbol{H}_m, \boldsymbol{x}_i)$ with the outputs of the model $m_{\theta^*}(\boldsymbol{x}_i)$.

**Neural Network $\pi_\psi$.** The inductive model is a three-layer fully-connected neural network with two Leaky ReLU activations and a sigmoid activation after the last layer. The input to $\pi_\psi$ is the concatenation $(\boldsymbol{H}_m; \boldsymbol{o}_{m,i}; \boldsymbol{x}_i; y_i)$. The hidden dimensions of the two intermediary layers are 64 and 16, and the final layer is a single neuron that outputs the score corresponding to a data point $\boldsymbol{x}_i$. While training $\pi_\psi$ we add a regularization term $\lambda'(\sum_{i \in D} \pi_\psi(\boldsymbol{H}_m, \boldsymbol{o}_{m,i}, \boldsymbol{x}_i, y_i) - |S|)$ to ensure that nearly $|S|$ samples have high scores out of the entire dataset $D$. Both the regularization constants $\lambda$ (in equation 3) and $\lambda'$ are set to 0.1. We train the model weights using an Adam optimizer with a learning rate of 0.001. During training, at each iteration we draw instances using $\text{Pr}_\pi$ and use the log-derivative trick to compute the gradient of the objective. During each computation step, we use one instance of the ranked list to compute the unbiased estimate of the objective in (3).

### D.4 Hyperparameter Optimization

The hyperparameter search was done over optimizer and scheduler-based hyperparameters using Ray Tune [29]. We set the possible optimizers to be SGD, Adam and RMSprop, and the possible schedulers to be CosineAnnealing and StepLR. The search space parameters are given below:

- *Optimizers*
  1. SGD: `learning_rate` $\in (0.001, 0.1)$, `momentum` $\in (0.7, 1.0)$, `weight_decay` $\in (0.01, 0.0001)$
  2. Adam: `learning_rate` $\in (0.001, 0.1)$, `weight_decay` $\in (0.01, 0.0001)$
  3. RMSprop: `learning_rate` $\in (0.001, 0.1)$, `momentum` $\in (0.7, 1.0)$, `weight_decay` $\in (0.01, 0.0001)$
- *Schedulers*
  1. StepLR: `step_size` $\in [10, 20, 30, 40]$, `gamma` $\in (0.05, 0.5)$
  2. CosineAnnealingLR

We employ TPE [2] as the hyperparameter search algorithm, and ASHA [28] as the hyperparameter scheduling algorithm. The hyperparameter search runs for 100 epochs in all cases. The random baseline is run for 5 runs, and we report the average speedup and test error.

# E   Additional experiments

## E.1   Comparison with additional baselines

Here, we compare the performance of SUBSELNET against two baselines. They are the two variants of our method–Bottom-$b$-loss and Bottom-$b$-loss+gumbel.

In Bottom-$b$-loss, we sort the data instances based on their predicted loss $\ell(g_\beta(\boldsymbol{H}_m, \boldsymbol{x}), y)$ and consider those points with the bottom $b$ values.

In Bottom-$b$-loss+gumbel, we add noise sampled from the gumbel distribution with $\mu_{\text{gumbel}} = 0$ and $\beta_{\text{gumbel}} = 0.025$, and sort the instances based on these noisy loss values, $i.e.$, $\ell(g_\beta(\boldsymbol{H}_m, \boldsymbol{x}), y) + \text{Gumbel}(0, \beta_{\text{gumbel}} = 0.025)$.

Figure 7 compares the performance of the variants of SUBSELNET, Bottom-$b$-loss, and Bottom-$b$-loss+gumbel. We observe that Bottom-$b$-loss and Bottom-$b$-loss+gumbel do not perform that well in spite of being efficient in terms of time and memory.

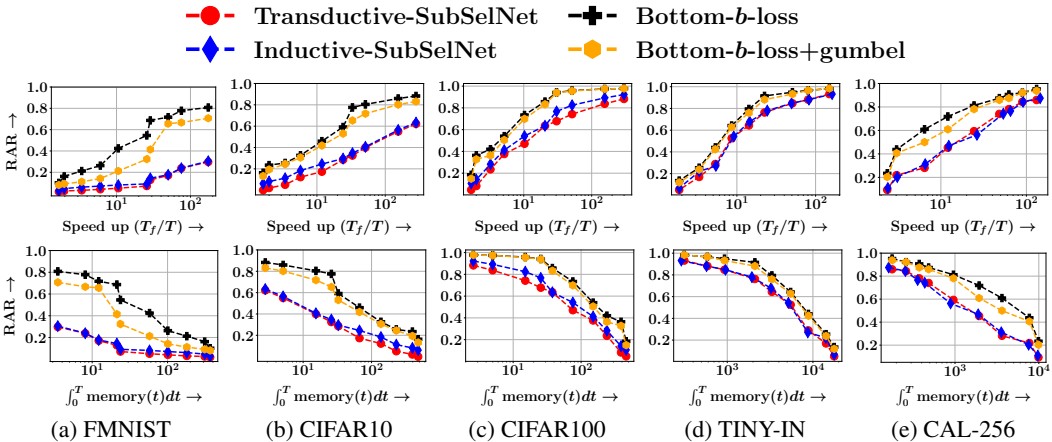

Figure 7: Comparison of Transductive-SUBSELNET and Inductive-SUBSELNET with Bottom-$b$-loss and Bottom-$b$-loss+gumbel. In Bottom-$b$-loss, we select top-$b$ instances in terms of their predicted loss $\ell(g_\beta(\boldsymbol{H}_m, \boldsymbol{x}), y)$ computed using the model approximator. In Bottom-$b$-loss+gumbel, we add gumbel noise $\text{Gumbel}(0, 0.025)$ to the loss and sort the instances based on these noisy loss values.

## E.2   Comparison of performance on ImageNet

Here, we compare the performance of SUBSELNET against two baselines: Selection-via-Proxy [5] and Pruning [48] on ImageNet-1K (1.28M images) [8] for $b \in (0.05|D|, 0.8|D|)$. Figure 8 compares Transductive-SUBSELNET and Inductive-SUBSELNET with the two baselines on basis of speedup, memory and budget.

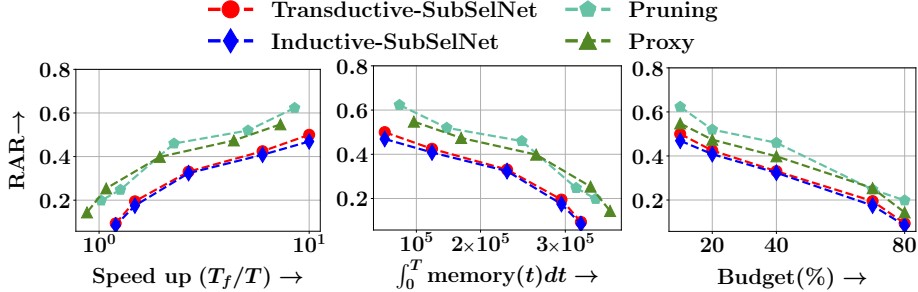

Figure 8: Comparison of performance of the top four non-adaptive subset selectors (Transductive-SUBSELNET, Inductive-SUBSELNET, Pruning, and Selection-via-Proxy) on ImageNet-1K [8] .

## E.3 Analysis of compute efficiency in high accuracy regime

We analyze the compute efficiency of all the methods, given an allowance of reaching 20% and 10% of the relative accuracy reduction (RAR) (80% and 90% of the accuracy) achieved by training on the full data. We make the following observations:

1. Transductive-SUBSELNET achieves the best speedup and consumes the least memory, followed by Inductive-SUBSELNET.
2. For CIFAR100, Tiny-Imagenet and Caltech-256, Bottom-$b$-loss and Bottom-$b$-loss+gumbel achieve better performance than the baselines which are able to reach the desired RAR milestones (10% or 20%).
3. For CIFAR100, Tiny-Imagenet and Caltech-256, most baselines could not achieve an accuracy of either 10% or even 20% of the RAR on full data.

| | FMNIST | | | | CIFAR10 | | | | CIFAR100 | | | |
|---|---|---|---|---|---|---|---|---|---|---|---|---|
| | Speedup | | Memory | | Speedup | | Memory | | Speedup | | Memory | |
| **Method** | 10% | 20% | 10% | 20% | 10% | 20% | 10% | 20% | 10% | 20% | 10% | 20% |
| GLISTER | 5.64 | 7.85 | 98.51 | 116.36 | 1.52 | 2.12 | 515.96 | 365.05 | 0.54 | 1.02 | 1427.77 | 758.55 |
| GradMatch | 4.17 | 5.24 | 136.40 | 243.75 | 1.69 | 2.20 | 457.67 | 362.47 | — | 0.84 | — | 917.04 |
| EL2N | 6.50 | 16.42 | 77.63 | 139.89 | 1.93 | 4.78 | 413.90 | 170.03 | — | — | — | — |
| GraNd | — | 1.18 | — | 450.73 | — | — | — | — | — | — | — | — |
| FacLoc | 0.82 | 2.37 | 652.67 | 81.01 | — | 0.80 | — | 558.56 | — | — | — | — |
| Pruning | 3.12 | 4.68 | 559.44 | 19.55 | 3.54 | 5.53 | 221.10 | 139.41 | — | 1.71 | — | 452.09 |
| Selection-via-Proxy | 3.65 | 18.09 | 168.20 | 35.27 | 1.95 | 1.03 | 819.22 | 410.26 | — | 1.02 | — | 765.05 |
| Bottom-$b$-loss | 1.68 | 2.98 | 393.28 | 190.40 | — | 1.77 | — | 433.07 | — | 1.67 | — | 465.39 |
| Bottom-$b$-loss+gumbel | 2.70 | 10.18 | 203.83 | 59.37 | 1.63 | 2.04 | 489.30 | 363.07 | — | 1.78 | — | 446.95 |
| Inductive-SUBSELNET | 28.64 | 69.24 | 22.73 | 8.24 | 3.63 | 8.99 | 221.99 | 99.54 | 1.93 | 2.82 | 417.16 | 274.17 |
| Transductive-SUBSELNET | 28.63 | 68.36 | 21.25 | 8.24 | 5.61 | 16.52 | 142.45 | 53.67 | 2.35 | 3.47 | 331.45 | 222.91 |

| | Tiny-Imagenet | | | | Caltech-256 | | | |
|---|---|---|---|---|---|---|---|---|
| | Speedup | | Memory | | Speedup | | Memory | |
| **Method** | 10% | 20% | 10% | 20% | 10% | 20% | 10% | 20% |
| GLISTER | 1.65 | 2.18 | 26705.5 | 22687.6 | 1.31 | 1.76 | 16904.5 | 13921.4 |
| GradMatch | 1.53 | 2.08 | 28249.9 | 25530.4 | 1.45 | 2.07 | 16499.3 | 12507.9 |
| EL2N | — | 2.30 | — | 21811.1 | — | — | — | — |
| GraNd | — | 2.16 | — | 24822.6 | — | — | — | — |
| FacLoc | — | — | — | — | — | — | — | — |
| Pruning | — | — | — | — | — | — | — | — |
| Selection-via-Proxy | — | 1.04 | — | 30717.2 | — | — | — | — |
| Bottom-$b$-loss | — | 2.54 | — | 15947.6 | — | — | — | — |
| Bottom-$b$-loss+gumbel | 1.88 | 2.72 | 17624.4 | 15085.9 | — | 2.36 | — | 9910.12 |
| Inductive-SUBSELNET | 2.02 | 3.54 | 17326.6 | 14505.4 | 2.43 | 3.16 | 9597.22 | 7406.23 |
| Transductive-SUBSELNET | 2.54 | 3.97 | 16281.1 | 12447.1 | 2.33 | 3.12 | 9983.86 | 7747.82 |

Table 8: Time and memory in reaching 10% and 20% RAR (90% and 80% of maximum accuracy of Full selection) in tradeoff curve in Figure 2 and 3 for all datasets. In the table, "—" denotes that under the current setup of experiments, i.e., the range of subsets considered, the method could not attain an accuracy equal to or less than 20% or 10% of RAR. Note that Bottom-$b$-loss and Bottom-$b$-loss+gumbel are variants/ablations of our method.

## E.4 Recommending model architecture

When dealing with a pool of architectures designed for the same task, choosing the correct architecture for the task might be a daunting task - since it is impractical to train all the architectures from scratch. In view of this problem, we show that training on smaller carefully chosen subsets might be beneficial for a quicker alternative to choosing the correct architectures. We first extract the top 15 best performing architectures $\mathcal{A}^*$ having highest accuracy, when trained on full data. We mark them as "gold". Then, we gather top 15 architectures $\mathcal{A}$ when trained on the subset provided by our models. Then, we compare $\mathcal{A}$ and $\mathcal{A}^*$ using the Kendall tau rank correlation coefficient (KTau) along with Jaccard coefficent $|\mathcal{A} \cap \mathcal{A}^*|/|\mathcal{A} \cup \mathcal{A}^*|$.

Figure 9 summarizes the results for three non-adaptive subset selectors in terms of the accuracy, namely - Transductive-SUBSELNET, Inductive-SUBSELNET and FL. We make the following ob-

servations: (1) One of our variant outperforms FL in most of the cases in CIFAR10 and CIFAR100. (2) There is no consistent winner between Transductive-SUBSELNET and Inductive-SUBSELNET, although Inductive-SUBSELNET outperforms both Transductive-SUBSELNET and FL consistently in CIFAR100 in terms of the Jaccard coefficient.

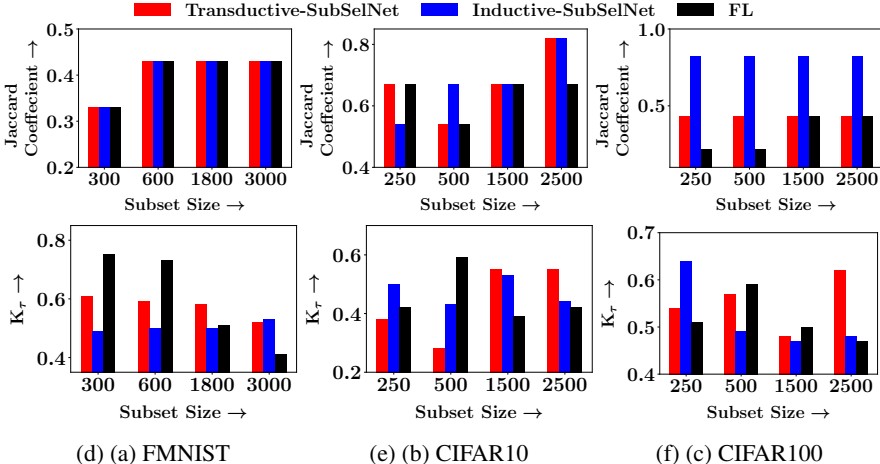

(d) (a) FMNIST          (e) (b) CIFAR10          (f) (c) CIFAR100

Figure 9: Comparison of the three non-adaptive subset selectors (Transductive-SUBSELNET, Inductive-SUBSELNET and FL) on ranking and choosing of the top-15 architectures on the basis of Jaccard Coefficient and Kendall tau rank correlation coefficient ($K_\tau$).

## E.5 Analysis of subset overlap on different architectures

|  | CIFAR10 | CIFAR100 |
|---|---|---|
| GLISTER | 0.05 | 0.06 |
| GRAD-MATCH | 0.06 | 0.08 |
| Our | 0.08 | 0.08 |

Table 9: Jaccard coefficient for subsets chosen by dissimilar architectures

Different architectures will produce different feature representations of the underlying dataset, and they can be distributed in different manners. Thus to generate a subset, if the features are different, we would expect subsets to change too. We experiment with extremely dissimilar architectures (top-5 ranked by distance in the latent space generated by the GNN) to observe the subset overlap occurring. Table 9 containing Jaccard coefficient of the subsets chosen for dissimilar architectures, where we notice that the overlaps are extremely small for the top adaptive methods, as well as for our method.

## E.6 Finer analysis of the inference time

|  | Transductive | Inductive | FL |
|---|---|---|---|
| Subset selection | 0.23 | 0.067 | 226.29 |
| Training | 70.1 | 70.1 | 70.1 |

Table 10: Inference time in seconds

Next, we demarcate the subset selection phase from the training phase of the test models on the selected subset during the inference time analysis. Table 10 summarizes the results for top three non-adaptive subset selection methods for $b = 0.005|D|$ on CIFAR100. We observe that: (1) the final training times of all three methods are roughly same; (2) the selection time for Transductive-SUBSELNET is significantly more than Inductive-SUBSELNET, although it remains extremely small as compared to the final training on the inferred subset; and, (3) the selection time of FL is large— as close as 323% of the training time.

## E.7 Analysis on underfitting and overfitting

Since the amount of training data is small, there is a possibility of overfitting. However, the coefficient $\lambda$ of the entropy regularizer $\lambda H(\mathrm{Pr}_\pi)$, can be increased to draw instances from the different regions

| Subset Size | Training | | Validation | | Testing | |
|---|---|---|---|---|---|---|
| | Transductive | Inductive | Transductive | Inductive | Transductive | Inductive |
| 10% | 0.728 | 0.660 | 0.702 | 0.632 | 0.678 | 0.606 |
| 20% | 0.852 | 0.673 | 0.809 | 0.658 | 0.770 | 0.644 |
| 40% | 0.890 | 0.691 | 0.856 | 0.678 | 0.825 | 0.666 |
| 70% | 0.942 | 0.738 | 0.912 | 0.717 | 0.884 | 0.698 |

Table 11: Variation of accuracy with subset size of both the variants of SUBSELNET on training, validation and test set of CIFAR10

of the feature space, which in turn can reduce the overfitting. In practice, we tuned $\lambda$ on the validation set to control such overfitting.

We present the accuracies on (training, validation, test) folds for both Transductive-SUBSELNET and Inductive-SUBSELNET in Table 11. We make the following observations:

1. From training to test, in most cases, the decrease in accuracy is $\sim 7\%$.
2. This small accuracy gap is further reduced from validation to test. Here, in most cases, the decrease in accuracy is $\sim 4\%$.

### E.8 Additional results on NAS and HPO

**Searched cell for NAS.** Figure 10 shows the final Normal and Reduction cells found on the DARTS search space using SUBSELNET on the 40% subset of CIFAR10, which gave the lowest test error of 2.68 in the experiments.

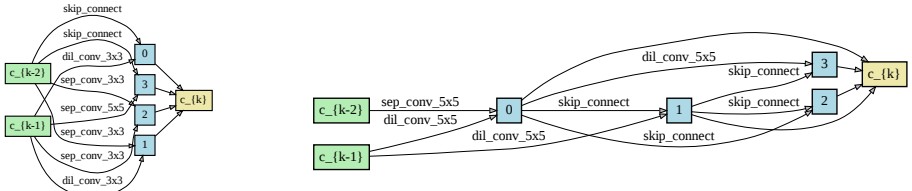

Figure 10: Normal [left] and Reduction [right] cells found by SUBSELNET during Neural Architecture Search using a 40% subset of the CIFAR10 dataset.

**Standard error results on NAS and HPO.** Here, we present the mean and standard error over the runs for the Test Error (%) on NAS and HPO, which has been presented in the main draft (Section 5.2). We observe our method offers less deviation across runs. Moreover, we found that the gain offered by our method is statistically significant with $p \approx 0.05$.

| Method | $b = 0.1|D|$ | $b = 0.2|D|$ | $b = 0.4|D|$ |
|---|---|---|---|
| Random | $3.02 \pm 0.171$ | $2.88 \pm 0.167$ | $2.96 \pm 0.169$ |
| Proxy-data [39] | $2.92 \pm 0.168$ | $2.87 \pm 0.167$ | $2.88 \pm 0.167$ |
| Our | $2.82 \pm 0.166$ | $2.76 \pm 0.164$ | $2.68 \pm 0.161$ |

Table 12: Mean and standard error of Test Error (%) on architectures given by NAS on CIFAR10

| Method | $b = 0.05|D|$ | $b = 0.1|D|$ |
|---|---|---|
| Random | $5.44 \pm 0.226$ | $3.72 \pm 0.189$ |
| AUTOMATA | $5.26 \pm 0.223$ | $3.39 \pm 0.181$ |
| Our | $4.11 \pm 0.199$ | $2.70 \pm 0.162$ |

Table 13: Mean and standard deviation of Test Error (%) for hyperparameters selected by HPO on CIFAR10

# F  Pros and cons of using GNNs

We have used a GNN in our model encoder to encode the architecture representations into an embedding. We chose a GNN for the task due to following reasons -

1. Message passing between the nodes (which may be the input, output, or any of the operations) allows us to generate embeddings that capture the contextual structural information of the node, i.e., the embedding of each node captures not only the operation for that node but also the operations preceding that node to a large extent.

To better illustrate the impact of the GNN, we compared it with a baseline where we directly fed the graph structure to the model approximator using the adjacency matrix, in lieu of the GNN-derived node embeddings. This alteration resulted in a notable performance decline, leading to a 5-6% RAR on subset size of 10% of CIFAR10.

| Variations of embedding | RAR | | KL-div |
|---|---|---|---|
| | $b = 0.05|D|$ | $b = 0.1|D|$ | |
| Feedforward $(g_\beta, \mathbf{A})$ | 0.481 | 0.433 | 0.231 |
| Feedforward $(g_\beta, \mathbf{H})$ | 0.434 | 0.397 | 0.171 |
| LSTM $(g_\beta, \mathbf{A})$ | 0.471 | 0.436 | 0.224 |
| LSTM $(g_\beta, \mathbf{H})$ | 0.412 | 0.386 | 0.102 |
| Attn. $(g_\beta, \mathbf{A})$ | 0.362 | 0.317 | 0.198 |
| Attn. $(g_\beta, \mathbf{H})$ | 0.310 | 0.260 | 0.089 |

Table 14: RAR and KL-div for different embeddings (**A**: Adjacency Matrix, **H**: GNN embedding) in model approximator

2. It has been shown by [38] and [57] that GNNs are as powerful as the Weisfeiler-Lehman algorithm and thus give a powerful representation for the graph. Thus, we obtain smooth embeddings of the nodes/edges that can effectively distill information from its neighborhood without significant compression.
3. GNNs embed model architecture into representations independent of the underlying dataset and the model parameters. This is because it operates on only the nodes and edges— the structure of the architecture and does not use the parameter values or input data.

However, the GNN faces the following drawbacks -

1. GNN uses a symmetric aggregator for message passing over node neighbors to ensure that the representation of any node should be invariant to a permutation of its neighbors. Such a symmetric aggregator renders it a low-pass filter, as shown in [41], which attenuates important high-frequency signals.
2. We are training one GNN using several architectures. This can lead to the insensitivity of the embedding to change in the architecture. In the context of model architecture, if we change the operation of one node in the architecture (either remove, add or change the operation), then the model's output can significantly change. However, the embedding of GNN may become immune to such changes, since the GNN is being trained over many architectures.

# G  Licensing Information

The NAS-Bench-101 dataset and DARTS Neural Architecture Search are publicly available under the Apache License. The baselines GRAD-MATCH and GLISTER, publicly available via CORDS, and the apricot library used for Facility Location, are under the MIT License.

