# OpenReview forum: "Efficient Data Subset Selection to Generalize Training Across Models: Transductive and Inductive Networks"
_NeurIPS.cc/2023/Conference — NeurIPS 2023 poster_

### Official Review · Reviewer_awrD · 2023-06-23

**Soundness:** 4 excellent
**Presentation:** 3 good
**Contribution:** 4 excellent
**Rating:** 9
**Confidence:** 2

**Summary:**

The authors train 1) an architecture encoder using a GNN with a variational graph autoencoder loss to learn a good representation for model architectures, 2) a model approximator taking the GNN representation and a sample x to predict the output of the model given x. The goal is to learn a probability distribution over the samples of the datasets to obtain a subset of the data with minimum loss while penalizing an entropic objective to encourage maximum diversity. Given a chosen architecture, Transductive-SUBSELNET directly optimizes all probabilities of the samples. Meanwhile, Inductive-SUBSELNET learns a neural network taking the dataset (and both the GNN representation and the model approximator output) to generate a probability for each sample. They also do a hybrid approach for better efficiency.

Although I understand the algorithm, this is quite complicated, and I cannot claim to understand all the specifics of every moving part (for example, how the GNN forms a representation for the architecture and learn to be insensitive to the model parameters, and why we need BFS; mainly I am not very familiar with GNNs).

**Strengths:**

The results are very in-depth, showing a lot of interesting ways to apply this general method and some useful applications like making NAS and hyperparameter-tuning nearly as good by much faster by reducing the dataset size. They present comparisons to other works, showing significantly improved performance in terms of RAR and memory. They train on many datasets of various image sizes but only in the image domain.

I think that this is great work.

**Weaknesses:**

Limited to CNNs for now (but that's okay). The biggest thing that could be improved is the notation and formatting of the paper to make the approach clearer, a lot of the details like in the GNNs could be left to the appendix. I feel like the paper overcomplicates things.

**Questions:**

Have you considered using pre-trained graph models for the architecture representation? It seems like there should exist some, and they may be trained with more than 250 architectures. If there are, it would be a worthwhile comparison and remove one big step from the algorithm. I know that there are GNNs for weights such as GHN-3, but I am not familiar with those for architecture.

It would be great to see b in Figure 3, maybe in the top x-axis, because right now, it's difficult to infer which size of the data is used.

You mention taking the images and then taking the 2048 Resnet-50 embedding as image representation; it seems to have come out of nowhere. Could you add more context to this? Because I assume that the actual images are used for real architectures and possibly in the Inductive-SUBSELNET neural network.

The notation can sometimes be weird, especially with the big dot in Pr_pi(dot); shouldn't x_i replace the dot?

I think that instead of separating algorithms 1 and 2 as training vs inference, it would be easier to understand by separating them into two algorithms with one Transductive and one Inductive, and each algo could be separated into top and bottom where the top is training and the bottom is inference. TRAINPIPELINE could be separate.


**Limitations:**

Yes, the authors properly address limitations and broader impacts.

---

> ### Author Rebuttal · Authors · 2023-08-10
>
> We thank the reviewer for the positive review of the paper!
>
> > *Have you considered using pre-trained graph models for the architecture representation? It seems like there should exist some, and they may be trained with more than 250 architectures. If there are, it would be a worthwhile comparison and remove one big step from the algorithm. I know that there are GNNs for weights such as GHN-3, but I am not familiar with those for architecture.*
>
>
> GNNs embed model architecture into representations independent of the underlying dataset
> and the model parameters (Appendix F page 9, L789). Thus we train GNN  once for all and the same GNN embeddings are used in all datasets. Indeed, this training was done at the very initial stage of the experiments and the resulting embeddings are being used everywhere.
>
> We did not find any pre-trained graph models for architecture representation. There are pre-trained graph embeddings for social network type graphs, i.e., Amazon recommendation network, open academic graph [1]; and molecular graphs [2]. However, we are not aware about similar pre-trained weights for architectures. If our work gets accepted, we will release the pre-trained graph embeddings, so that it can be directly used for subsequent work without further training from scratch.
>
> [1] Hu et al. GPT-GNN: Generative Pre-Training of Graph Neural Networks, KDD 2020.
>
> [2] Xia et al.  Mole-BERT: Rethinking Pre-training Graph Neural Networks for Molecules, ICLR 2023
>
>
>
> > *It would be great to see b in Figure 3, maybe in the top x-axis, because right now, it's difficult to infer which size of the data is used.*
>
> Following the suggestion, we have added the budget to the computational efficiency plots in the top x-axis in Figure 1, and as a separate plot in Figure 2 in the global-pdf.
>
>
> > *You mention taking the images and then taking the 2048 Resnet-50 embedding as image representation; it seems to have come out of nowhere. Could you add more context to this? Because I assume that the actual images are used for real architectures and possibly in the Inductive-SUBSELNET neural network.*
>
>
> We sincerely apologize for the misunderstanding which arose in Section 5.1 about usage of the ResNet-based embedding for images. To calculate the similarity matrix for facility location, we utilize the penultimate-layer of the ResNet-based feature extractor to represent the image representation, which is common in literature especially for calculating similarity between images. Hence, the feature extractor information on L276 is used **only during the subset selection stage** for the facility location formulation in L281. The input to the model architectures for all the methods and $\texttt{SubSelNet}$ neural network are the actual images.
>
>
>
> > *The notation can sometimes be weird, especially with the big dot in Pr _pi(dot); shouldn't x _i replace the dot?*
>
> We use $\Pr _\pi(\bullet)$ as the distribution itself, which is over the subset $S$. Here, $S$ can replace the dot.
>
>
> > *I think that instead of separating algorithms 1 and 2 as training vs inference, it would be easier to understand by separating them into two algorithms with one Transductive and one Inductive, and each algo could be separated into top and bottom where the top is training and the bottom is inference. TRAINPIPELINE could be separate.*
>
> We thank the reviewer for this suggestion - we will update the presentation algorithm and split it separately into Transductive and Inductive.

---

> > ### Comment · Reviewer_awrD · 2023-08-16
> >
> > I'm very satisfied with the reviewer response, I am not changing my score of 9/10.

---

### Official Review · Reviewer_U6Yc · 2023-07-06

**Soundness:** 3 good
**Presentation:** 3 good
**Contribution:** 3 good
**Rating:** 6
**Confidence:** 2

**Summary:**

Current subset selection methods are architecture specific and requires solving an optimization problem for each architecture individually. The subset selected through solving an optimization for one architecture does not generalize to another model. This paper addresses this problem and introduces an end-to-end training method using multiple architectures (through embedding them via GNN) to select a generalizable subset. The paper addresses an important problem, written thoroughly and consists of sufficient empirical experiments to verify the idea.

**Strengths:**

- Addresses a relevant practical problem in selecting subsets which can generalize across different architectures.
- The paper is generally very well written and each component of the method pipeline is very well explained.
- Good speedup with the proposed design (which is very relevant in practical scenarios) and also lower RAR compared to other baselines.

**Weaknesses:**

- While the method is a generalization of the subset selection algorithm — it is inherently complex which can lead to less adoption by the community
- The paper will be stronger if more complex datasets are added. Currently only smaller datasets are used, which can obfuscate the real-world capabilities of this method. While Tiny-ImageNet is used, a few experiments at ImageNet scale will help verify the efficacy of the method more strongly.
- I would expect a little bit more analysis on the characteristics of the data subset selected using their method vs. other baselines. Are there difference in the properties of data which are selected by SubSelNet? What is the overlap factor with other methods?

**Questions:**

See Weakness

**Limitations:**

Overall, I believe that the paper is well-written with good empirical results. Given that the method (although not simple for adoption) offers a significant speedup for new architectures, I would vote for a weak acceptance.

---

> ### Author Rebuttal · Authors · 2023-08-10
>
> We thank the reviewer for their detailed reading of the paper and their positive feedback.
>
>
> > *The paper will be stronger if more complex datasets are added. Currently only smaller datasets are used, which can obfuscate the real-world capabilities of this method. While Tiny-ImageNet is used, a few experiments at ImageNet scale will help verify the efficacy of the method more strongly.*
>
>
> We want to note that Tiny-ImageNet (a 100,000 point subset of ImageNet with 200 classes), and CalTech-256 (a 30,607 point dataset with 257 classes) are challenging real-world datasets, especially for subset selection. We further conducted experiments on ImageNet and presented the results in Figure 2 of the global-pdf and also here in the following table which shows the compute time and memory to reach 10% and 20% RAR values. We observe that we perform better than other methods (- means that we couldn’t achieve that RAR with the method).
>
>
> |  | Speedup |  | Memory (Gb-min) |  |
> |---|---|---|---|---|
> | **RAR** $\to$ | 10\% | 20\% | 10\% | 20\% |
> | Pruning | 0.99 | 1.03 | 7.31e5 | 6.94e5 |
> | Random | 1.06 | 1.21 | 6.90e5 | 5.69e5 |
> | Proxy | 0.91 | 0.99 | 8.24e5 | 7.19e5 |
> | Our | 1.25 | 1.47 | 5.77e5 | 4.87e5 |
>
>
>
> > *I would expect a little bit more analysis on the characteristics of the data subset selected using their method vs. other baselines. Are there difference in the properties of data which are selected by SubSelNet? What is the overlap factor with other methods?*
>
> Note that the key to our algorithm is to select different subsets that are optimal to different architectures. In that way, we found that the similarity between architectures has positive correlation between subsets.
> We compute similarity between graphs $s(G _i,G _j)$  and the underlying subsets $|S _i\cap S _j|$ for each pair of architectures $i,j$. Then we get the Kendall’s-tau (k) between the list of all possible pairs of $s(G _i,G _j)$ and $|S _i\cap S _j|$ to be 0.42 for CIFAR10 and 0.55 for CIFAR100. This shows that there is a positive correlation between the model structure and the subset chosen.
>
> We also observed overlap between subsets returned by other methods with ours. Indeed there is a good amount of overlap. For example, we observe that for CIFAR100, the subset chosen by us has a 31% overlap with the subset chosen by GRAD-MATCH. However, it is to note that our method selects the subset significantly faster than others, as shown in Table 4 of the global-pdf.

---

> > ### Comment · Reviewer_U6Yc · 2023-08-16
> > **Response to authors**
> >
> > I thank the authors for their detailed response; I will maintain my rating!

---

### Official Review · Reviewer_VPxr · 2023-07-08

**Soundness:** 3 good
**Presentation:** 3 good
**Contribution:** 2 fair
**Rating:** 6
**Confidence:** 3

**Summary:**

This work proposes a new method to select subsets of valuable training examples, with an emphasis on specializing the selections to new model architectures. The motivation is that selections made for one architecture may not work well when used with a different architecture (a claim that the authors did not show evidence for here). At inference time, the method takes a new network architecture, converts it to a learned embedding via a GNN, feeds it along with new inputs $x$ to a model approximator that estimates predictions from a trained model, and then optimizes over a probability vector from which we can sample a valuable subset. Rather than training the different components all at once, the authors propose separate objectives for each learned module. And intuitively, the final optimization over the probability vector (shown in eq. 1) encourages low loss among the selections, as well as high diversity.

**Strengths:**

This is the first method I've seen that focuses on specializing data subset selection to specific architectures, so that angle is novel. It's also challenging, but the authors manage this complexity by setting up a GNN and model approximator to efficiently estimate predictions from trained models with arbitrary architectures. The full pipeline is involves multiple steps, but the authors show how to train each module independently and the results show that it works well in practice.

**Weaknesses:**

- One of the main motivators for this method is the idea that data subset selection cannot generalize across architectures. I didn't see any evidence for this claim in the paper, and in fact there are other works that show the opposite conclusion: for example, the selection via proxy approach from Coleman et al (2020) relies on the generalization of valuable data from smaller to larger models. Can the authors explain their focus on this issue? And if they believe it is a serious issue, can they provide evidence of lack of transferability across models, and perhaps of very different selections in their method depending on the architecture used at inference time?

- The authors provide no information about the computational cost of the pre-processing step in this method. If the goal is to perform efficient neural architecture search or hyperparameter tuning, this cost should be accounted for, and it seems to require training at least 250 models with different architectures, plus the GNN and model approximator. Ultimately, it's hard to say if this actually reduces the computational cost of training.

- The terminology for "transductive" and "inductive" solutions for $\pi$ didn't make sense to me. I wonder if there's a simpler way to refer to these methods - for example, the "transductive" method solves a per-instance optimization problem, and the "inductive" method can be viewed as an amortized optimization solution (see "Tutorial on Amortized Optimization" by Amos).

- The main objective shown in eq. 1 didn't quite make sense to me. The first part of the objective, which focuses on the loss for the selected examples, seems like it wouldn't capture anything meaningful - many samples will be perfectly predicted and have near-zero loss, and it doesn't seem especially helpful to focus on the easiest examples that are correctly predicted. Am I missing something, or can the authors explain this potential issue?

- It would seem like the diversity term in the eq. 1 objective is intended to ensure diversity in some representation space and minimize redundancy for very similar training examples. Indeed, that's what prior methods aim to accomplish with submodular set functions like facility location. So the choice to instead use the entropy over $P_\pi$ is strange to me, because it cannot be expected to encourage this diversity. In fact, I'm not sure what it accomplishes because we're already performing sampling without replacement. Can the authors explain this choice, and explain whether it would be possible to test their method with a submodular set function here instead of the entropy term?

- I'm not sure why the authors included eq. 3 given that they don't suggest training all the models end-to-end like this. It's not clear that this end-to-end objective would even result in good models for the GNN and model approximator. I wonder if the authors could skip this somewhat confusing overview objective and proceed directly to the training steps for each module.

- In section 4.1 about the GNN, could the authors confirm which architectural information they encode in the input representation? It sounded like they only indicate the operation performed at each layer, but perhaps not the size of each layer. If so, that seems somewhat limiting for the GNN's ability to distinguish between different architectures.

- The main text doesn't seem to explain how the authors optimize eq. 9 and eq. 10. We optimizing over $\pi$ here (or a network that outputs $\pi$), and this would seem to require backpropagating gradients to a large number of logits through a non-differentiable sampling process; in particular, the "inductive" solution seems like it could require a forward pass through all the training examples for each gradient step. Algorithms 1 and 2 provide no information, and I didn't see a pointer to an appendix section that explains this point.

- The notation for the "inductive" model on line 235 is confusing because it doesn't show the inputs referenced in the same sentence. This is only corrected below on line 241.

- The RAR metric is sensible, but it also leads to a y-axis in Figure 3 that's difficult to interpret. In particular, it's hard to tell at what point the trained models have unacceptably low accuracy due to too much speedup. Would the authors consider instead showing raw accuracy, or absolute accuracy reduction?

- I wonder if the authors could include strong but simple baselines to compare to the existing ones. For example, could they include training on the full dataset for different numbers of epochs, training on a fixed random subset, or training on newly sampled subsets at each epoch? Given the focus on speedup on the x-axis rather than the number of epochs, I could see these baselines outperforming some of the existing ones, which are in some cases quite slow.

**Questions:**

Several questions are mentioned in the issue above.

One of the baselines used here, EL2N, was recently shown to have a serious bug in its implementation that invalidated some of the findings. Can the authors explain the relevance of that issue to their experiments and say whether they're using a corrected implementation here?

**Limitations:**

The paper does not include much discussion of limitations.

---

> ### Author Rebuttal · Authors · 2023-08-10
>
> Thanks for detailed feedback which would help us improve our paper.
>
> > *evidence of lack of transferability*
>
> We provided a comparison with Selection via proxy in Fig 7 of App E.1 (also Fig 3, global-pdf), where we outperform it. Also, we select 5% subsets of CIFAR10 for 4 architectures with the lowest no. of parameters and use them to train another architecture. We observe that this simple baseline gives RAR of (0.36, 0.39, 0.37, 0.37) whereas our method gives RAR=0.31. Thus, the subset obtained from one architecture may not be the best for other architectures.
>
> >  *cost of pre-training*
>
> GNN is trained only once, irrespective of datasets since it only captures the structure of the architecture. The model approximator indeed involves an overhead, but it is one-time and is amortized by no. of unseen architectures during inference time training.
> To give an analogy, premier search engines invest a lot of resources in making fast inferences rather than training (e.g., LLM). They build complex models that are difficult and computationally intensive, but their inference is fast for several queries once trained. We have discussed them in detail in Limitations, App.A.
>
> We have provided this amortization analysis in Table 6 (L387) of our main paper. Here, we show that even when we include training times into total cost, we still gain a better speedup (CIFAR10).
>
> |# of archs for training approximator|100||||250||||
> |-|-|-|-|-|-|-|-|-|
> ||Speedup||Memory (Gb-min)||Speedup||Memory (Gb-min)||
> |RAR $\to$|10%|20%|10%|20%|10%|20%|10%|20%|
> |GLISTER|1.52|2.12|515.96|365.05|1.52|2.12|515.96|365.05|
> |GradMatch|1.69|2.20|457.67|362.47|1.69|2.20|457.67|362.47|
> |Inductive|3.18|5.23|253.40|171.10|2.94|4.41|274.08|202.91|
> |Transductive|4.25|7.21|188.03|122.97|3.65|4.97|218.94|178.39||
>
> Note that the model approximator and pre-trained models can be trained only once for a large dataset and be fine-tuned for smaller datasets. This allows us to significantly reduce total computation cost at least by a factor of 5, with nearly same performance.
>
> > *New metric: absolute accuracy reduction, Simple baselines:  fixed random subset, newly sampled subsets at each epoch*
>
> We added new plots in Fig 1 of the global-pdf and show that SubSelNet performs better
>
> > *main objective in eq. 1*
>
> Note that the diversity term allows it to select examples from different areas of the instance space, by adding several hard examples. At the outset, one can think of the problem as $\min\sum  _{i \in S}\ell(m  _{\theta}(x _i),y _i)$ s.t. $\text{Diversity(S)}\ge a$.
>
> Thus, we aim to choose a representative subset $S^*$ among many subsets $S$, which gives minimum training error. As $S^*$ is a representative set, most test examples are likely to be in the regime of S. The trained model $\hat{\theta} (S^*)$ gives the lowest error across all other models trained across different representative subsets.  Therefore, it gives lowest loss on most test examples, since they will be fall in the regime of $S^*$.  Indeed, there are some hard examples giving bad accuracy, which, however, is significantly compensated by very high accuracy of most test examples. Explicit training over hard examples leads the model to learn unnecessary on outliers, resulting in overall accuracy drop (26% for CIFAR10). $a$ ($\lambda$ in our original objective) controls this regularization.
>
> Generalization of subset selection is already very challenging. As the model is trained on diverse and easy examples, the neural pipeline finds it easy to generalize this task across architectures. In contrast, selecting hard examples is already quite difficult for one architecture. This hardness gets amplified when we try to generalize these across different architectures.
>
> > *$P _\pi$ is strange*
>
> We tried to approximate Facility location (FL) using a neural network. For a non-sequential $Pr _\pi$, we have: $\sum  _{j\in D} \max _{i\in S} sim _{ij} \approx \sum _{j\in D} \log(\sum  _{i\in D}  e^{sim  _{ij}} Pr _\pi (i) ) $. We faced several difficulties here. (1) We need to pass all examples twice, so it can be computationally expensive $O(|D|^2)$.  As $|S|>1$, $\sum _{i} Pr _{\pi} (i) > 1$ – we cannot relax it with Jensen inequality. (2) The sequential extension when sampling of one instance will affect subsequent sampling is very non-trivial to incorporate here. (3) $sim _{ij}$ is computed in some latent feature space that requires architecture embeddings, model approximator, etc. and thus it demands another neural network on top of all current networks, which would further complicate the training.
>
>
> Hence, we choose to increase entropy of $P _{\pi}$ at each selection step. This leads the softmax probability distribution (Eq 8) across candidate instances to be close to uniform at each step. This will allow us to choose samples uniformly from different regimes. This will lead to choosing a representative subset although we agree that it would not encourage minimum redundancy.  We saw that in practice this works very well, and also allows for log-derivative tricks to perform efficient sampling.
>
> > *What does GNN encode*
>
> The entire neural architecture is fed to GNN including all layer (thus including model size), structural and operation information.
>
> > *optimize eq. 9 and eq. 10 over $\pi$*
>
> We use log derivative trick to compute $\nabla _{\psi} \mathbb{E}  _{S\sim P  _{\pi _\psi}}[ \Lambda (S,\psi) ] = \mathbb{E}   _{S\sim P  _{\pi _\psi}}[ \nabla  _{\psi}\log P  _{\pi _\psi} (S)\Lambda(S,\psi)+ \nabla _ {\psi} \Lambda(S,\psi) ]$. This would allow us to (1) compute  gradient for backpropagation and (2) distribute product of softmax probabilities (Eq 8 in the paper) into sum of log probabilities, which allows to compute outer expectation easily.
>
> > *El2N bug*
>
> We believe to be using the correct (updated) version of the code, where the bug in flax was fixed in April 2021, regarding the loading of checkpoints.
>
> > *lack of limitations*
>
> We discussed limitations in detail in App A.

---

> > ### Comment · Reviewer_VPxr · 2023-08-12
> > **Thanks for response**
> >
> > Thanks to the authors for their response. I remain skeptical of several design choices in this work (the need for an architecture encoder, the wisdom of the main objective, the entropy penalty which is apparently chosen for convenience), and like some other reviewers I find it overly complicated and highly burdensome for users. Regarding the lack of transferability across architectures, the response offered by the authors is not exactly convincing: their results are slightly better than SVP, but it's not as if SVP doesn't work at all due the smaller architecture, which seems implied by the paper's narrative.
> >
> > The redeeming factor here is that the results are in fact better. I'm not enough of an expert in these methods to have a strong view on why, but the authors deserve credit for their results. Given my concerns, I plan to keep my score as "borderline accept." I would be interested to chat with the other reviewers at some point.

---

> > > ### Author Response · Authors · 2023-08-12
> > > **Clarifying some of the concerns**
> > >
> > > Many thanks for your quick response.  We would like to address your concern, esp. for the justification between neural pipeline and SVP once more. We would earnestly request you to have a look into it. Given the character limit, we could  not delve into details of design choices especially architecture encoder in the original rebuttal. However, we describe them in details here.
> > >
> > >
> > > **SVP**: We would like to clarify that the improvement over SVP is indeed significant over our method. We apologize for the miscommunication. The numbers  (0.36, 0.39, 0.37, 0.37)  we showed in the following statement
> > >
> > > “We observe that this simple baseline gives RAR of (0.36, 0.39, 0.37, 0.37) whereas our method gives RAR=0.31. Thus, the subset obtained from one architecture may not be the best for other architectures. “
> > >
> > > are NOT the results for selection of proxy. Here, we use *our method* to compute the optimal subset  and use it train *only one new architecture*. Neither, they are results from SVP nor are they averaged across architectures. We used this example to simply show that optimal subsets using our method may differ across architectures and the intention was not to contrast/compare SVP against our method.
> > > We simply used one randomly drawn architecture to show that the optimal subsets may differ. This difference  is actually large if we aggregate across architectures.
> > >
> > > In fact, in Figure 3 of the rebuttal PDF, the plots indeed show that the results of our method are indeed significantly better than SVP,  for complex datasets like TinyImagenet and Caltech 256.
> > >
> > > For example in TinyImagenet,  RAR values of **our method  is 0.52,  SVP is 0.71** for 20% subset,  and for caltech 256,   **ours is 0.46 and SVP is 0.67.**
> > >
> > > We believe that it does show that our method is overall significantly better SVP, which showcases that subsets trained on one architecture may not generalise across others.
> > >
> > >
> > > **Explanation on the flow of our pipeline:**  Generalization of the subset selection across architectures is an extremely complex task. Since we are generalizing the process across architectures, in a non-modelAgnostic manner, the pipeline becomes a bit complex.  When we generalize across any objects, we always convert them into embeddings, so that vectorial operations can be performed. Since the architectures are the objects over which we wish to generalize, we convert them into vector. For graph structured objects, GNN are the state-of-the-art for this purpose.  We made a detailed discussion in the response to Reviewer KRgz. We reproduce it here once again for your convenience.  Here, we try to establish a reasoning behind using a transformer-based network along with the architecture encoder for model approximation.
> > >
> > >
> > > First note that the architectures under consideration can be represented as directed acyclic graphs, with forward message passing. During the forward computation, at any layer for node $v$, the output $a(v)$ can be represented as
> > >
> > > $$a{(v)}=H _v\left(\sum  _{u\in InNbr(v)} op  _v(a(u)) \right) \quad \text{with} \ a (\text{root}) = x----- (A)$$
> > >
> > > Here, $op _v$ is the operation on the inputs coming into the node. E.g., $op _v$ can be simply a linear matrix multiplication and $H _v$ is the activation function at node v.
> > > We are interested in $a (\text{OutNode})$ where OutNode is the final node where the output is computed. Now, given the nature of this recursion,  a graph neural network--- which exactly operates like above--- can approximate $a  (\text{OutNode})$ with appropriate nonlinearities.
> > >
> > > Specifically, GNN will gather messages from $k = 1,...,K$ hop starting with with $h _0 = nodeFeature$ as follows:
> > > $$h  _{k+1} (v)=NN_1\left(\sum  _{u \in InNbr(v)}NN _2(h   _{k+1}(u)) \right)$$
> > >
> > > Here, $NN_1$ and $NN_2$ are neural networks. Since GNN operates exactly as the computation process (A), it makes sense to assume that $h _K(1),...,h  _K(V)$ are good representations of the operation within the architecture. They, together with the feature $x$ should be able to predict the $a (\text{outNode})$.
> > >
> > > Thus, our task is now to find nonlinearities $F$ and $G$ so that
> > >
> > > $a(\text{outNode}) \approx  F( G (h _K(1), …, h _K(|V|)), x )$.
> > >
> > > Now, the set $(h _K(1), …, h _K(|V|))$ is permutation equivariant with respect to node indexing.
> > > As suggested in [1], transformers are universal approximators of permutation equivariant functions. Therefore we use G as a transformer. Furthermore, we apply another neural network F on top of output of G and x to predict $a$(outNode).
> > >
> > > [1] Yun et al.  Are Transformers universal approximators of sequence-to-sequence functions? ArXiv, abs/1912.10077.
> > >
> > > We believe that the above reasoning justifies our design choices to some extent. Note that the problem is extremely challenging and largely unaddressed in literature. As a first step, some of the design choices may not be rigorously justified but they showed consistent good results across datasets; and thus, contributes to a crucial step forward.

---

> > > > ### Comment · Reviewer_VPxr · 2023-08-14
> > > > **Thanks for clarification**
> > > >
> > > > Thanks for the clarification, I can see that the accuracy/speed-up trade-off vs SVP is indeed favorable. This work would be more valuable to the community if it showed comprehensive evidence that data subset selection depends strongly on the architecture; this is currently treated as a premise (which contradicts the literature) and there are only a few results supporting the point. I'm willing to raise my score to "weak accept," but I think the paper could be improved through a revision, and I remain negative about the overall complexity and burden for using this method.

---

> > > > > ### Author Response · Authors · 2023-08-21
> > > > >
> > > > > We thank the reviewer for the reply and increasing their score!
> > > > >
> > > > > > *This work would be more valuable to the community if it showed comprehensive evidence that data subset selection depends strongly on the architecture; this is currently treated as a premise (which contradicts the literature) and there are only a few results supporting the point.*
> > > > >
> > > > > Different architectures will produce different feature representations of the underlying dataset, and they can be distributed in different manners. Thus to generate a subset, if the features are different, we would expect subsets to change too.
> > > > > We experiment with extremely dissimilar architectures (top-5 ranked by distance in the latent space generated by the GNN) to observe the subset overlap occurring. We present the results below in the table containing Jaccard coefficient of the subsets chosen for dissimilar architectures (higher value means high set similarity)–
> > > > >
> > > > > |  | GLISTER | GRAD-MATCH | Our |
> > > > >  |:---:|:---:|:---:|:---:|
> > > > >  | CIFAR10 | 0.05 | 0.06 | 0.08 |
> > > > > | CIFAR100 | 0.06 | 0.08 | 0.08 |
> > > > >
> > > > > We notice that the Jaccard coefficients are very low across different datasets and methods, showing that subsets might vary between architectures.

---

### Official Review · Reviewer_X3mh · 2023-07-08

**Soundness:** 2 fair
**Presentation:** 1 poor
**Contribution:** 3 good
**Rating:** 5
**Confidence:** 3

**Summary:**

The paper presents a model agnostic method of subset selection using a graph neural network a surrogate.

**Strengths:**

- Problem area is an interesting space, and an area of interest to the community at present.
- Novel application of GNN for subset selection
- Seems to be a performance gain in some settings for SubSelNet over existing methods (but hard to tell from graphs).

**Weaknesses:**

- The paper not well written. I found the paper hard to parse. Structure is messy, and the paper visually feels like there's too much going on. The figures are small. The graphs and tables are hard to read. This paper would not have a much of an impact as it stands, simply because it's difficult to read.
- Doesn't seem to a convincing narrative here on why are existing methods not model agnostic? See Selection via Proxy by Coleman et al and RHO-Loss by Mindermann et al.
- Datasets are all small. The graphs are hard to read
- The method seems overly complicated and some of the justifications seems slim (e.g. GNN for architecture encoder? why architecture encoder? Why "neural approximator"?


Minor Comments:
- The arrows (on RAR as other metrics) in figure 3 and table 1 are confusing. They don't indicate in the way arrows usually do (i.e. what direction is best). I don't even know why there is an arrow the table 1.

**Questions:**

See above.

But on the whole this paper is just needs a thorough restructuring at the very least.

**Limitations:**

Societal Impact: Would be nice to see the effect of subset selection on fairness.

---

> ### Author Rebuttal · Authors · 2023-08-10
>
> We thank the reviewer for their comments.
>
> > *The paper presents a model agnostic method of subset selection using a graph neural network a surrogate.*
>
> We believe that there is a misunderstanding. Our method is not model agnostic. Rather, it chooses a different subset for each architecture using a neural model which is trained to select optimal subsets given an architecture. This allows us to obtain a subset that is optimal, specifically for a given architecture, without any combinatorial or other expensive subset selection algorithm.
>
> Thus, although it does not require to explicitly train any model or solve a combinatorial optimization problem every time, it selects a subset using a neural subset selector, which is indeed different for different architectures.
>
> > *Doesn't seem to a convincing narrative here on why are existing methods not model agnostic?
>
> In the introduction, we mentioned that many subset selection methods that show good accuracy are required to train everytime from scratch for every new architecture. This is time consuming.
> Indeed  there are works like Selection via Proxy by Coleman et al which selects subsets efficiently, without training the model again and again. However, they perform worse in terms of accuracy with respect to those methods which attempt to select a subset each time for a new architecture. This is natural because the model specific subset selectors tailor their subset selection per each architecture.
>
> Our method attempts to effectively tradeoff between these two aspects. By training on several architectures, we aim to learn to select a subset that is optimal for a new architecture, without explicit training or expensive combinatorial computation.
>
> Note that we already compared Selection via proxy by Coleman et al (2020) in Figure 7 (Appendix E.1), which shows that our approach provides a better trade-off than their method. For convenience, we have shown the plot in Fig 3 of the global-pdf. Moreover,  we show the compute time and memory to reach 10% and 20% RAR values for FMNIST, and observe that our method outperforms others.
>
> ||Speedup||Memory (Gb-min)||
> |--|:-:|-:|:-:|:-:|
> ||10\%|20\%|10\%|20\%|
> |Selection by Proxy|3.65|18.09|168.20|35.27|
> |Inductive|28.64|69.24|22.73|8.24|
> |Transductive|28.63|68.36|21.25|8.24|
>
> To further enhance the fact that the subsets to be chosen depend not only on the dataset but also on the architecture under consideration, we selected 5% subsets from 4 architectures with the lowest number of parameters and used them to train a larger architecture. We observe that this simple baseline gives RAR - (0.36, 0.39, 0.37, 0.37) whereas our method gives RAR of 0.31.
>
> > *Datasets are all small*
>
> We want to note that Tiny-ImageNet (a 100,000 point subset of ImageNet with 200 classes), and CalTech-256 (a 30,507 point dataset with 257 classes) are challenging real-world datasets, especially for subset selection. We further conducted experiments on ImageNet and presented the results in Fig 2 and also here in the following table which shows the compute time and memory to reach 10% and 20% RAR values. We observe that we perform better than other methods.
>
> |  | Speedup |  | Memory (Gb-min) |  |
> |---|---|---|---|---|
> |**RAR** $\to$ | 10\% | 20\%|10\% | 20\% |
> |Pruning|0.99| 1.03 | 7.31e5|6.94e5 |
> |Random|1.06 |1.21 | 6.90e5|5.69e5 |
> |Proxy|0.91| 0.99 | 8.24e5|7.19e5 |
> |Our|1.25|1.47 | 5.77e5|4.87e5 |
>
> > *The method seems overly complicated and some of the justifications seems slim (e.g. GNN for architecture encoder? why architecture encoder? Why "neural approximator"?*
>
> Our task is as follows: Given an architecture m and a training D, we should be able to find a subset S *without explicitly training m*, which would give optimal performance across all subsets of the same size. That means we need an algorithm A which would take input m and D and will output S, *without any explicit training of m on D*.  Thus, the goal is to find A such that $A(m,D)=S^* _m$, the optimal subset for the architecture m.
>
> In general, such an algorithm should solve some candidate optimization problem (Eq 1 in the paper) like the following:
>
> $$\min _{\theta, S\subset D: |S|=b}\sum _{i \in S}\ell(m _\theta(x _i),y _i)-\lambda\texttt{Diversity}(\{x _i|i\in S\})$$
>
> *Why a neural approximator?:* The key bottleneck here is to train $m_\theta(x)$ every time for a new architecture. Our work focuses on bypassing this expensive step by providing an approximation of the *trained output of $m_\theta (x)$*, without explicit training of $m_\theta$.  Hence, whenever we need to train $m_\theta$, we can use the approximation directly which will replace the entire training stage.
>
> To do so, we design a neural approximator which takes the architecture of m and the instance x as input and provides the output $m_{\theta^*} (x)$, the prediction made on x by the trained architecture.  Whenever we require to train the model m during the subset selection procedure, we directly feed the architecture of m and x into the neural approximator and directly obtain the output $m_{\theta^*} (x)$ — the entire process of training $m_\theta$ is removed.
>
> Once we have the approximation of the trained model output, we feed it into another neural network, which directly samples the subset.
>
> *Why architecture encoder?*  The neural approximator aims to make efficient generalization of predicting model output across different architectures. We note that generalizing any task across the different architectures requires the architectures to be embedded in vector space. However, directly using the graph matrices doesn’t encompass the operations or structure completely, as shown in Table 1 in the global pdf.
>
> > *The arrows in RAR*
>
> The $\uparrow$ in Figure 3 denotes the axis, and $\to$ in Table 1 of the main paper denotes that **RAR** refers to 10% and 20%. We will address them in the revised version.

---

> > ### Comment · Reviewer_X3mh · 2023-08-18
> > **Response**
> >
> > I've now read the other reviews, along with the authors rebuttal.
> >
> > Responses to my concerns firstly.
> >
> > | We believe that there is a misunderstanding. Our method is not model agnostic.
> >
> > No misunderstanding. I meant model agnostic from the perspective that it can be applied to any model architecture rather than the data selected can be used to trained any model architecture.
> >
> > | In the introduction, we mentioned that many subset selection methods that show good accuracy are required to train everytime from scratch for every new architecture. This is time consuming. Indeed there are works like Selection via Proxy by Coleman et al which selects subsets efficiently, without training the model again and again.
> >
> > Okay but then what about adaptive methods like RHO-LOSS. Considering the setting, this seems like the most appropriate baseline then considering the majority of the method tested are non-adaptive?
> >
> > | ImageNet Results and Results more generally.
> >
> > Firstly, question for all the results, this seems like one single run was conducted for each experiment. In my experience, these methods tend to have non-trivial standard-deviations and at the very least 3 runs would be necessary. Apologies for not raising this earlier, it was on my notes on the paper, but I seem to have missed adding it to the review.
> > Secondly, the gains here seem small. 1.21 using random subset vs 1.41. Including standard deviations, I would say this difference might be even smaller. Add in the pretraining time, this would be negligible.
> >
> > | On the justifications for the components
> >
> > I understand that. Equation 1 is a reasonable thing to optimise for. The part I felt wasn't justified was each component working. e.g. neural approximator. Upon re-reading the paper, I do think the KLs are helpful here. I would be nice have a baseline for this (KL between two of the same models with different initialisations, and perhaps ImageNet pre-trained ResNet or something). However, this does resolve some of my concerns here.
> >
> > | readability
> >
> > This paper is still hard to parse, and I saw no acknowledgments of that throughout this rebuttal. The rebuttal response further provided further evidence of that. The figures, graphs, tables and spacing make the very aesthetically unappealing and just difficult to read. I wouldn't vote for a rejection for this on it's own but it is a contributing factor. This paper has the potential to be impactful, but only if someone reads it. The paper would really benefit from additional time here.

---

> > > ### Author Response · Authors · 2023-08-20
> > > **Clarification of the concerns**
> > >
> > > Many thanks for your response. We attempt to clarify your concerns once again. We would earnestly request you to look into it.
> > >
> > > > *RHO-LOSS…the majority of the methods are non-adaptive*
> > >
> > > We have tested our method against 4 adaptive baselines, and 2 non-adaptive baselines (described in L279 of the paper).  Here, we compare our method against RHO loss. We observe that we outperform RHO loss by a significant margin in CIFAR10.
> > > ||Speedup||Memory (Gb-min)||
> > > |-|-|-|-|-|
> > > ||10\% RAR|20\% RAR|10\% RAR|20\% RAR|
> > > |RHO-Loss|1.09|1.37|390.3|310.26|
> > > |Our|5.61|16.52|142.45|53.67|
> > >
> > > > *ImageNet Results and Results more generally.*
> > >
> > > No. of runs: Due to some baselines being computationally very expensive, we couldn’t run them multiple times for multiple architectures (we indicated this in the openreview submission form). Although we could have reported standard errors except those baselines, we thought that such partial reporting would be confusing and further given that there are so many points in the graph, error bars would make it look more difficult to parse.
> > >
> > > In general, we observed that **seeding doesn’t affect the pareto efficiency overall**. In the table below, we report the speedup and memory, along with their standard errors for 10 runs, required to reach 10% to 40% RAR for the most efficient baselines. The gap for others is even larger.
> > >
> > > (Speedup)
> > > ||10% RAR|20% RAR|30% RAR|
> > > |-|-|-|-|
> > > |Pruning|3.54±0.06|5.53±0.12|7.97±0.22|
> > > |EL2N|1.93±0.03|4.78±0.11|7.34±0.11|
> > > |Our|5.61±0.06|16.52±0.18|28.36±0.41|
> > >
> > > (Memory (GB-min))
> > > ||10% RAR|20% RAR|30% RAR|
> > > |-|-|-|-|
> > > |Pruning|221.1±3.75|139.41±3.03|87.13±2.41|
> > > |EL2N|413.9±6.43|170.03±3.91|101.88±1.53|
> > > |Our|142.45±1.52|53.67±0.58|29.18±0.42|
> > >
> > > ImageNet:  Achieving 10% or 20% RAR requires |S| = 0.88|D| and |S| = 0.81|D| for random whereas, we reach this with |S| = 0.76|D| and 0.63|D|. Despite this larger size for random,  we had to draw several random subsets, to ensure at least one lucky random subset gives  10% or 20% RAR. We reported the speedup/memory for only that subset, without incorporating how many subsets thus generated could not reach the desired RAR. Hence, the probability that the speedup taken by random is 1.21 is very small.
> > >
> > > For fair comparison, we now compute $\mathbb{E}  _S[\text{time taken}|RAR(S)=\text{Desired RAR}]$, by generating different sized-subsets, which reach required RAR.  We notice there is a significant gap between our and random.
> > >
> > > ||10% RAR|20% RAR|30% RAR|40% RAR|
> > > |-|-|-|-|-|
> > > |Random|1.02±0.03|1.06±0.06|1.38±0.06|1.93±0.08|
> > > |Our|1.25±0.03|1.47±0.03|2.47±0.04|6.02±0.07|
> > >
> > > Further, we also calculate the difference in ours and random’s performance for a fixed speedup, where we notice that we have a 18.15% and 21.08% RAR gain when targeting a 5x or 10x speedup, which are significant.
> > >
> > > > *KL with pre-trained ResNet*
> > >
> > > We had also considered pre-trained ResNet during the initial stage of experiments, since it can significantly reduce the pre-training cost. However, the performance was poorer, with a KL of 0.261, whereas our method achieves a KL of 0.089.
> > >
> > > > *paper is hard to parse, I saw no acknowledgment*
> > >
> > > We sincerely apologize for not addressing this in the response. We are thankful to the reviewer to point essential aesthetic and readability changes out in the paper, especially to the plots and tables. However the character limit during the rebuttal was 6k and we thought of giving the overall explanation of our method more priority there.
> > >
> > > We compared the variants of our method with  6 baselines in main, and 3 more in the appendix. This generated complex trade-off plots, which made the graphs difficult to parse. On the other hand, generating a single large table out of so many numbers might be overwhelming, and we decided to keep the figures. On top of that, for quick reference to subparts of our method along with ablations, we aimed to keep those tables in the main paper for quick access by the reader. However, we understand that this made results difficult to parse
> > >
> > > Please note that NeurIPS provides an additional page for accepted papers. If our paper gets accepted, we would use this for presentation changes in the paper, as follows:
> > > - We will split the large main figure into two larger figures (separating adaptive and non-adaptive baselines), so that every plot is clearly readable. We will also put FMNIST in the Appendix, so that the figure in the main can be made larger.
> > > - We will fix the notations and structures in the table, to make the results more apparent without being a source of confusion.
> > > - To reduce notational overheads, we will move the low-level GNN description to appendix and describe the theoretical underpinning.
> > > - We will additionally place SVP and RHO-Loss in the introduction and the method section, to motivate them better.
> > >
> > > Rebuttal:  We got 6 reviews and we were required to report many results. Thus, unfortunately we crowded the global pdf with results. However, if the paper gets accepted, we will distribute them in the same manner.

---

> > > > ### Comment · Reviewer_X3mh · 2023-08-20
> > > > **Thanks for the comprehensive response.**
> > > >
> > > > Thanks for the response. Most of my concerns resolved. I've changed my rating to 5.

---

> > > > > ### Author Response · Authors · 2023-08-21
> > > > >
> > > > > We thank the reviewer for increasing the score!

---

### Official Review · Reviewer_rt8F · 2023-07-23

**Soundness:** 2 fair
**Presentation:** 3 good
**Contribution:** 3 good
**Rating:** 5
**Confidence:** 3

**Summary:**

The paper proposes a new subset selection method, SUBSELNET, for efficient training of neural network models. SUBSELNET selects optimal subsets for a model based on its architecture with minimal overhead. SUBSELNET outperforms existing subset selection methods in terms of computational efficiency.

**Strengths:**

New data-driven approach to subset selection: the proposed method turns subset selection into a learning problem and learns a subset selection pipeline that can generalize to new architectures. Such approach has the potential of being more universal while also more efficient at certain conditions (see below).

A step towards uncovering the relationship between optimal subsets and model architecture: the proposed method in principle shows that optimal subsets can be determined from a simple representation of the model architecture, which is a quite strong hypothesis worth investigating. This line of research has the potential to better our understanding of the relationship between data and model in deep learning.

Efficient subset selection adaptive to mode architecture: although with significant drawbacks (see below), the proposed method points to a promising possibility of architecture-adaptive subset selection with very little cost.

Application in NAS and hyperparameter search seem promising: from the results reported in the paper, the proposed method has the potential to significantly accelerate NAS and hyperparameter search if used properly.

**Weaknesses:**

Method has limited efficiency advantage because of costly training: the proposed method needs to train a large number of models in order to learn a model approximator, therefore though it is more efficient than other adaptive subset method at test time, the computation cost is considerably shifted to the training phase. As a result, the proposed method is only clearly more efficient when it is required to efficiently train a large number of models of different architectures so that cost can be amortized. This situation seems only happen when doing NAS. NAS is undeniably an important task, but otherwise the potential use of the proposed method seem limited.

(minor) Another limitation of the method is that the trained model approximator does not generalize across datasets. The method needs to be retrained for each new dataset, which further limits its scope of application. The method also requires a pre-trained feature extractor suitable for the current data distribution, which may not be available for less common datasets.

Insufficient analysis of the components in the proposed method: to appreciate the merits of the proposed design of approximating model prediction based on architecture, one would want to know how effective is the model approximator. For example, how effective does it generalize to unseen architectures? Also, to what extent is the subset selection dependent on architecture (how much do subsets selected for two different architectures overlap, and does the degree of overlap depend on architecture similarity)? For the GNN architecture encoder, how much does it contribute to the final performance? And what about the pre-trained feature extractor?

(minor) Scope of experiment is kind of limited: experiment is mainly performed on small datasets such as CIFAR100, no experiment on larger datasets such as ImageNet where efficient training is much more meaningful.

Some assumptions made in the paper are likely questionable: see the `Questions` section.

Potentially incomplete evaluation protocol: see the `Questions` section.

**Questions:**

Assumptions on architecture encoder:

Section 4.1 describes the parameterization of operations in a neural network into node representations, and use them as features in the model approximator to approximate the prediction of a model. This implicitly assumes that the prediction of a model depends only on the structure of the network, which is likely not the case. Hyperparameters like model size, optimizer, learning rate, etc., can all affect model performance and could affect different architectures differently. It may require empirical analysis to show if using architecture alone suffices to approximate model predictions.

Assumptions on using SUBSELNET in AutoML:

The idea of using adaptive subset selection in NAS, etc., is potentially problematic. The goal of NAS is to select $\underset{m}{\operatorname{argmin}} L$, while the proposed method selects $\underset{m}{\operatorname{argmin}} \underset{S}{\operatorname{min}} L$ which is a different optimization problem. Because the subset can adapt to each particular model, the performance on the model's most advantageous subset might not correlate well with its performance on the whole dataset. For example, it is possible that a generally worse-performing model has a good training subset. On the other hand, non-adaptive subset selection does not risk over-adaptation between subset and model and can be more safely used in NAS to approximate $\underset{m}{\operatorname{argmin}} L$.

Evaluation:

The main results of the paper, Figure 3 and Table 1, are mostly dedicated to comparing the speed (computation time) and memory usage. Here the computation time involves multiple components (e.g., running subset selection, model training) and can be tricky to compare across methods and is implementation-dependent. Because the proposed method uses costly training, to include the amortize training time or not in the comparison could make a large difference. The authors need to be more explicit in the detailed protocol used when comparing computation time.

Because the training phase involves training many models, the cost is mostly linked to the number of models trained. Therefore, it seems necessary to verify the minimum number of models need to be trained in order to learn a good model approximator.

Comparison of efficient training can be evaluated in other dimensions as well, most notably with sample efficiency: would the proposed method be able to select a smaller subset to maintain certain accuracy? Sample efficiency is also a more reliable metric than computation time and allows for more objective comparison.

Results in "using SUBSELNET in AutoML" section: why only `random` and `proxy` are evaluated in Table 4, what about other subset selection methods listed in Figure 3? Why are non-adaptive baselines not evaluated in Table 6, which could have low cost?

**Limitations:**

The authors adequately addressed the main limitations of the proposed method, including dependence on the training dataset and the inability of generalizing model prediction to the test set.

---

> ### Author Rebuttal · Authors · 2023-08-10
>
> Thank you very much for your thorough reading and valuable feedback. Please find our answers below:
>
> > *how effective does model approximator generalize to unseen architectures?*
>
> Ablation study (L337) in our paper provides an evaluation on model approximators. Further, the analysis from L349 and Table 2 addresses this exact question. We evaluate the model approximator $g  _{\beta}$ alone---  without the presence of the subset sampler--- using KL divergence between the gold model outputs $m  _{\theta^*} (x   _i)$ and predicted model outputs $g   _{\beta}(H   _m,x   _i))$.  Here we measure $AVG  _{m \in M  _{test}} KL(m  _{\theta^*} (x   _i) || g  _{\beta}(H   _m,x   _i))$. For convenience, we reproduce the numbers from table 2 for CIFAR10.
> || KLDiv|
> |-|-|
> |FF|0.171|
> |LSTM|0.102|
> |Our|0.089|
>
> Further, we note that the classification accuracy of the model approximator  on unseen architectures taken from NASBench-101 is 96%, and from DARTS is 94% in CIFAR10
>
> > *to what extent is subset selection dependent on architecture?*
>
> As per the reviewer’s suggestion, we compute similarities between graphs $s(G _i,G _j)$ and the underlying subsets $|S _i\cap S _j|$ for each pair of architectures $i,j$. We calculate the Kendall’s-tau between the list of all possible pairs of $s(G _i,G _j)$ and $|S _i\cap S _j|$ to be 0.42 for CIFAR10 and 0.55 for CIFAR100. This shows that there is a positive correlation between the model structure and the subset chosen.
> Additional results in App 7 and Fig 3 in the global-pdf show that existing works, which assume subset for one model can generalize across others, are outperformed by our method.
>
> > *How much GNN contributes to the final performance*
>
> If we replace GNN embeddings with simple architecture embeddings, we see a 5-6% increase in RAR. We have added the details in the global response (GR.4) and Table 1 of the global-pdf.
>
> > *what about the pre-trained feature extractor?*
>
>  We have clarified about the extractor in the global response (GR.3)
>
>
>  > *Results on large dataset*
>
> Fig 2 in global-pdf and (GR.5) in the global response compares subset selection across baselines on ImageNet, which show the efficacy of our method.
>
>
> > *Hyperparameters like model size, optimizer, learning rate, etc., can all affect model performance and could affect different architectures differently*
>
> GNN encodes structure of the architecture and thus  *does incorporate model size*. Since other hyperparameters (H) are not structure related, we did not include them in the GNN. We also did not incorporate H in the model approximator as its performance as well as the final accuracy deviates by max 2-3% for different H and we noted that they mostly change the rate of convergence rather than the final accuracy for *all methods including training with full subset*. Thus, our plots (speedup/memory v RAR) remain almost the same.
>
>
> > *idea of using adaptive subset selection in NAS, etc., is potentially problematic*
>
> Note that, due to the entropy term (Eq 2), we are minimizing the loss over all representative subsets (over the full dataset). So, the optimal S is itself a representative subset which gives minimum loss. Hence, the model optimized on the subset in NAS also performs well in the full dataset. This is why we observe that this approach works well in practice.
>
> >  *(a) Computation time can be tricky to compare across methods; (b) Amortize training time or not in the comparison make a large difference; (c) the detailed protocol used when comparing computation time;*
>
> (a)  Whenever available, we used the available code of the baselines. All these baselines are specifically optimized for compute efficiency and therefore they did run in their full strength. Every experiment is on the same GPU server in exactly the same setting. During time/memory computation, we ensured that no other processes were running on the server.
>
> (b) Computation time including amortized training is analyzed in (GR.1) in the global-response which shows that our method still stays efficient.
>
> (c) As we mentioned in the paper (L306), “Computational efficiency…  $T$ is the time taken for the entire inference task, which is the average time for selecting subsets across the test models $m'\in \mathcal{M} _{test}$ plus the average training time of these test models on the respective selected subsets. “ Hence, this is total time = running subset selection+model training. Comparison on these two aspects separately is given in Tab 10 in the Appendix and Table 4 in the global-pdf.
>
>
> > *verify the minimum number of models need to be trained in order to learn a good model approximator.*
>
> We trained with different numbers of models $n$ and observed that beyond $n_{min}= 100$, we are able to train a good approximator. (Please refer to Table 3 in the global-pdf).
>
>
> > *Sample efficiency is also a more reliable metric than computation time and allows for more objective comparison.*
>
> We plot the accuracy-budget tradeoff in Fig 2 of the global-pdf for Imagenet, which shows that our method performs better than the other. Moreover, we also provide a table here which shows the value of $|S|$ required to reach 10–30% drop in max performance for ImageNet, where we notice that we are able to select a smaller subset to achieve the same RAR.
> **RAR**$\to$|10\%|20\%|30\%
> ---|---|---|---
> Pruning|0.86|0.80|0.62
> Random|0.88|0.81|0.69
> Proxy|0.83|0.74|0.56
> Our|0.76|0.63|0.44
>
>
> > *why only random and proxy in AutoML?*
>
> Random and proxy are the state-of-the-art specifically for NAS. During the rebuttal period, we also experimented with our non-adaptive baselines: Pruning and Facility location. We notice that our methods still selects better architectures for NAS. We put the results in the global-pdf in Table 2

---

> > ### Comment · Reviewer_rt8F · 2023-08-17
> >
> > The reviewer thank the authors for providing many extra results and clarifications.
> >
> > Most of my questions are answered, although I still do not feel extremely confident of the results and the generalizability of the findings given the complexity of the method itself and the complexity of the design settings that could affect the results. I will raise my score to the positive side.

---

> > > ### Author Response · Authors · 2023-08-21
> > >
> > > We thank the reviewer for their reply and increasing their score!
> > >
> > > > *I still do not feel extremely confident of the results and the generalizability of the findings given the complexity of the method itself and the complexity of the design settings that could affect the results.*
> > >
> > > In the common response titled *Why we use GNN and transformer: Insights from theoretical underpinning*, we have highlighted the construction through which we have decided to use the specific elements in the pipeline in their current form.
> > >
> > > Note that our method performed well across six diverse datasets. Moreover, through extensive ablation studies present in the main paper (Table 2, Line 337), Appendix E (Table 9, L663) and the global-pdf (Table 1 and 3), we have observed that although the GNN+Transformer blocks provide the best performance. Moreover, our design choice is robust.  For example, we used a transformer as a sequence encoder in a model approximator. Here, if we use another sequence encoder like LSTM, we still outperform the nearest baseline (selection via proxy) by 9.1% and 7% RAR at subset sizes of 5% and 10%.  However, we observed that the transformer gives more performance boost.

---

### Official Review · Reviewer_KRgz · 2023-07-26

**Soundness:** 2 fair
**Presentation:** 3 good
**Contribution:** 3 good
**Rating:** 6
**Confidence:** 4

**Summary:**

This paper focuses on the Subset Selection problem for selecting the best subsets training samples with which a well-performed model can be trained. This paper focuses on the challenge that previous methods for subsets selection cannot transfer across architectures. Specifically, this paper proposes to use a transformer to predict the accuracy of a model trained on a subset of training samples and use the predicted results as the surrogate to solve the combinatorial optimization problems for the selection (Transductive). This paper also proposes inductive selection, which learns another neural to predict the result of combinatorial optimization. Further, this paper proposes to encode the neural network's architecture using GNN, which will also be fed to the predictor.

__________________________________________________ Post Rebuttal _________________________________________________

The reviewer has already read the rebuttal.

All my concerns have been addressed. And the reviewer wishes that the insights discussed in the rebuttal can be accommodated in the final version of the paper.

The reviewer will maintain the original score since the overall novelty, contribution, and impact cannot reach the reviewer's bar for papers scoring 7, and a score of 6 is already positive for the paper.

**Strengths:**

1. In this manuscript, the authors present an innovative approach toward subset selection, putting forward a new framework that emphasizes efficiency and practical applicability.

2. A feature of the proposed framework is its ability to generalize across various architectures. This is a step forward, as it ensures the versatility and universal applicability of the method. The framework is not only confined to a specific architectural layout, but it can adapt to and function effectively within a range of scenarios.

3. Additionally, the paper is substantiated by robust experimental results that highlight the superiority of the proposed method in comparison to existing baselines. The authors provide strong empirical evidence of the efficacy of their approach, demonstrating that it not only meets the performance standards set by previous methods but often surpasses them.

Minor:
The authors have also shown how their method can be naturally incorporated into Automated Machine Learning (AutoML) systems. This seamless integration is beneficial, as it minimizes compatibility issues that could arise when introducing a new method into an established system. AutoML stands to gain from this method as it could optimize the automated selection, deployment, and tuning of machine learning models, and the paper illustrates this potential effectively.

**Weaknesses:**

1. Despite the promising results presented in the manuscript, one key concern that emerges is the absence of a justification for the methods put forward. Specifically, it is not well-established whether the transformer in use is genuinely capturing the intricate mapping between models and data, or if it is merely overfitting to the training samples. Overfitting would lead to a lack of generalizability in unseen or novel data situations, which could severely limit the practical utility of the proposed method. It would greatly strengthen the paper if the authors could provide deeper insights into these theoretical underpinnings, perhaps by conducting additional analysis or testing to definitively establish the veracity of the transformer's operations.

2. Another potential shortcoming that emerges from the work is the limitation inherent in the utilized dataset. The dataset, NAS-Bench-101, contains Convolutional Neural Network (CNN) models with architectures that bear high similarities. Specifically, it is not clear whether the proposed method can be extended to other network architectures, such as Residual Networks (ResNets). It would therefore be beneficial if the authors could test their method with a broader range of architectures and provide evidence to support its efficacy across such a diverse range.

**Questions:**

To the best of the reviewer's knowledge, analysis over accuracy is an NP-hard problem. It remains elusive to the reviewer that transformer really has the ability to capture the mapping between neural networks and accuracy.

**Limitations:**

The proposed methods rely significantly on the pertaining of the model approximation to produce the correct accuracy and largely depend on the transferability of the model approximator.

---

> ### Author Rebuttal · Authors · 2023-08-10
>
> We thank the reviewer for the insightful feedback. We in turn answer the questions below:
>
> > *Despite the promising results presented in the manuscript, one key concern that emerges is the absence of a justification for the methods put forward. Specifically, it is not well-established whether the transformer in use is genuinely capturing the intricate mapping between models and data, or if it is merely overfitting to the training samples. It would greatly strengthen the paper if the authors could provide deeper insights into these theoretical underpinnings, perhaps by conducting additional analysis or testing to definitively establish the veracity of the transformer's operations. [...] To the best of the reviewer's knowledge, analysis over accuracy is an NP-hard problem. It remains elusive to the reviewer that transformer really has the ability to capture the mapping between neural networks and accuracy.*
>
> We would like to point out that the neural approximator captures the probability distribution across the classes — and does not directly predict the accuracy based on the features.  Note that the task is to choose the subset from the training set and therefore the values of gold labels are available to us. Since the labels are already given to us, they can be directly compared with the predicted outputs from the model approximator to provide the predicted accuracy. Hence, the model approximator does not aim to predict accuracy only based on the features— it simply predicts the model outputs and compares them with the gold labels.
>
> Next, we try to establish a reasoning behind using a transformer-based network along with the architecture encoder for model approximation. Given the hardness of the problem, we will leave the exact formal statement for future work.
>
> First note that the architectures under consideration can be represented as directed acyclic graphs, with forward message passing. During the forward computation, at any layer for node $v$, the output $a(v)$ can be represented as
>
> $$a{(v)}=H _v\left(\sum  _{u\in InNbr(v)} op  _v(a(u)) \right) \quad \text{with} \ a (\text{root}) = x----- (A)$$
>
> Here, $op _v$ is the operation on the inputs coming into the node. E.g., $op _v$ can be simply a linear matrix multiplication and $H _v$ is the activation function at node v.
> We are interested in $a (\text{OutNode})$ where OutNode is the final node where the output is computed. Now, given the nature of this recursion,  a graph neural network--- which exactly operates like above--- can approximate $a  (\text{OutNode})$ with appropriate nonlinearities.
>
> Specifically, GNN will gather messages from $k = 1,...,K$ hop starting with with $h _0 = nodeFeature$ as follows:
> $$h  _{k+1} (v)=NN_1\left(\sum  _{u \in InNbr(v)}NN _2(h   _{k+1}(u)) \right)$$
>
> Here, $NN_1$ and $NN_2$ are neural networks. Since GNN operates exactly as the computation process (A), it makes sense to assume that $h _K(1),...,h  _K(V)$ are good representations of the operation within the architecture. They, together with the feature $x$ should be able to predict the $a (\text{outNode})$.
>
> Thus, our task is now to find nonlinearities $F$ and $G$ so that
>
> $a(\text{outNode}) \approx  F( G (h _K(1), …, h _K(|V|)), x )$.
>
> Now, the set $(h _K(1), …, h _K(|V|))$ is permutation equivariant with respect to node indexing.
> As suggested in [1], transformers are universal approximators of permutation equivariant functions. Therefore we use G as a transformer. Furthermore, we apply another neural network F on top of output of G and x to predict $a$(outNode).
>
> [1] Yun, C., Bhojanapalli, S., Rawat, A.S., Reddi, S.J., & Kumar, S. (2019). Are Transformers universal approximators of sequence-to-sequence functions? ArXiv, abs/1912.10077.
>
> We leave the exact theory for future work.
>
>
>
> > *Another potential shortcoming that emerges from the work is the limitation inherent in the utilized dataset. The dataset, NAS-Bench-101, contains Convolutional Neural Network (CNN) models with architectures that bear high similarities. Specifically, it is not clear whether the proposed method can be extended to other network architectures, such as Residual Networks (ResNets). It would therefore be beneficial if the authors could test their method with a broader range of architectures and provide evidence to support its efficacy across such a diverse range*
>
> We want to note that the multi-stage training approach can be easily extended to architectures beyond the NAS-Bench search space. In section 5, we have also tested the methods on the DARTS space, and observed that our method outperforms other methods in terms of speed and computation efficiency. As suggested by the reviewer, we add the final results for ResNet below,  which shows the speedup and memory to achieve 10% and 20% RAR values across different methods for the top competitive baselines and observe that we outperform them in both aspects.
>
>
> ||Speedup|| Memory (Gb-min)||
> |--|:--:|--:|:-:|--:|
> | **RAR** $\to$ |  10\% |  20\% |   10\% |   20\%
> | GLISTER| 1.54 | 2.71 | 84.41 |  40.80 |
> | GradMatch | 1.44 | 3.36 | 79.74 | 35.47 |
> | Inductive | 3.57 | 10.31 | 23.30 |  8.07 |
> | Transductive | 3.61 | 10.35 | 23.04 | 8.03

---

> > ### Comment · Reviewer_KRgz · 2023-08-12
> > **Thanks for the well-prepared rebuttal**
> >
> > The reviewer wishes to thank the author for the well-prepared rebuttal.
> >
> > All my concerns have been addressed. And the reviewer wishes that the insights discussed above can be accommodated in the final version of the paper.
> >
> > The reviewer will maintain the original score since the overall contribution and impact cannot reach the reviewer's bar for scoring 7 papers and a score of 6 is already positive to the paper.

---

### Author Rebuttal · Authors · 2023-08-10

We thank all the reviewers for their constructive and insightful comments. We would like to summarize the reviews and the global-pdf (attached with this rebuttal) here.

> (GR.1) *Results after addition of pre-training time*

Currently, the approximator is trained with 250 architectures. However, we observed that even when we train the approximator with a lesser number of models, we beat the top competitive baselines after accounting for amortization. Below, we show the amortized time and memory to reach 10% and 20% RAR for the CIFAR10 dataset with the model approximator training architecture set of 100 and 250, for the most competitive methods.
|Approximator training size|100||||250||||
|---|---|---|---|---|---|---|---|---|
||Speedup||Memory||Speedup||Memory||
|**RAR**$\to$|10%|20%|10%|20%|10%|20%|10%|20%|
|GLISTER|1.52|2.12|515.96|365.05|1.52|2.12|515.96|365.05|
|Grad-Match|1.69|2.20|457.67|362.47|1.69|2.20|457.67|362.47|
|Inductive|3.18|5.23|253.40|171.10|2.94|4.41|274.08|202.91|
|Transductive|4.25|7.21|188.03|122.97|3.65|4.97|218.94|178.39||
> (GR.2)  *Other applications beyond NAS*

In addition to NAS, our work can be used for hyperparameter selection in the AutoML domain.

Primarily, we address the optimization of network-related hyperparameters such as activation functions and intermediate layer widths. The approach we propose involves training these model instances on a subset of data derived from our method. This expedited model training strategy quickly yields trained models and facilitates efficient cross-validation procedures.

Moreover, let us consider the case where we need to tune non-network hyperparameters, such as learning rate, momentum, and weight decay. Given the architecture, we can choose the subset obtained using our method to train the underlying model parameters for different hyperparameters, which can then be used for cross-validation.

> (GR.3) *Clarification regarding the pre-trained feature extractor*

We sincerely apologize for the misunderstanding which arose in Section 5.1 about usage of the ResNet-based embedding for images. To calculate the similarity matrix for facility location, we utilize the penultimate-layer of the ResNet-based feature extractor to represent the image representation, which is common in literature especially for calculating similarity between images. Hence, the feature extractor information on L276 is used **only during the subset selection stage for facility location** as shown in L281.
Thus, the input to the model architectures for $\texttt{SubSelNet}$ and other methods are the actual images.


> (GR.4) *Contribution of the GNN-based architecture encoder*

Discussion regarding the GNN is present in Appendix F.  Summarizing (1) GNN provides contextual embeddings of each node that captures not only the operation for that node but also the operations preceding that node. (2) GNN gives embeddings that are independent of the underlying dataset allowing us to train the encoder only once and use it for multiple datasets.
As a baseline, we feed the graph structure explicitly into the model approximator using the adjacency matrix instead of the GNN-derived node-embeddings. We note that such a change negatively impacts the performance, resulting in a 5-6% drop in RAR for 10\% subset on CIFAR10 as shown in Table 1 of the global-pdf.

> (GR.5) *Imagenet results*

We have added the results for ImageNet in Fig 2 in the global-pdf. We want to note that the adaptive baselines gave an out-of-memory error, and hence we were not able to experiment on those. Here we report the compute time and memory required to reach 10% and 20% RAR.

| | Speedup || Memory |  |
|---|---|---|---|---|
| **RAR** $\to$| 10\% | 20\% | 10\% | 20\% |
| Pruning | 0.99|1.03 | 7.31e5 | 6.94e5 |
| Random | 1.06 |1.21 | 6.90e5 | 5.69e5 |
| Proxy | 0.91 | 0.99 | 8.24e5 | 7.19e5 |
| Our | 1.25 | 1.47 | 5.77e5 | 4.87e5 |

> (GR.6) Summary of the global-pdf

In the global-pdf, we have added:

**Figure 1**: Tradeoff curves between absolute accuracy reduction (difference of test error of full selection and subset) and Speedup/Memory for the baselines: Random, Random-switch (random subset chosen at each epoch) and Selection-via-Proxy

**Figure 2**: Tradeoff curves between RAR and Speedup/Memory/Budget for ImageNet for the baselines: Random, Random-switch and Selection-via-Proxy

**Figure 3**: Tradeoff curves between RAR and Speedup for Tiny-ImageNet and Caltech-256 for the baseline: Selection-via-proxy

**Table 1**: Comparison of performance of GNN-based embedding and graph matrix embedding in the model approximator

**Table 2**: Test Error of NAS on DARTS search space for all non-adaptive baselines on CIFAR10

**Table 3**: Variation of SubSelNet performance on changing number of training architectures in the model approximator

**Table 4**: Breakdown of selection and training time for top adaptive and non-adaptive baselines on CIFAR100 for 0.5\% dataset

---

### Author Response · Authors · 2023-08-13
**Why we use GNN and transformer: Insights from theoretical underpinning**

Here, we try to establish a reasoning behind using a transformer-based network along with the architecture encoder for model approximation.  We already mentioned it in the response to reviewer Krgz, since the reviewer specifically asked for this.
For the convenience of all the reviewers, we further present it here.

First note that the architectures under consideration can be represented as directed acyclic graphs, with forward message passing. During the forward computation, at any layer for node $v$, the output $a(v)$ can be represented as

$$a{(v)}=H _v\left(\sum  _{u\in InNbr(v)} op  _v(a(u)) \right) \quad \text{with} \ a (\text{root}) = x----- (A)$$

Here, $op _v$ is the operation on the inputs coming into the node. E.g., $op _v$ can be simply a linear matrix multiplication and $H _v$ is the activation function at node v.
We are interested in $a (\text{OutNode})$ where OutNode is the final node where the output is computed. Now, given the nature of this recursion,  a graph neural network--- which exactly operates like above--- can approximate $a  (\text{OutNode})$ with appropriate nonlinearities.

Specifically, GNN will gather messages from $k = 1,...,K$ hop starting with with $h _0 = nodeFeature$ as follows:
$$h  _{k+1} (v)=NN_1\left(\sum  _{u \in InNbr(v)}NN _2(h   _{k+1}(u)) \right)$$

Here, $NN_1$ and $NN_2$ are neural networks. Since GNN operates exactly as the computation process (A), it makes sense to assume that $h _K(1),...,h  _K(V)$ are good representations of the operation within the architecture. They, together with the feature $x$ should be able to predict the $a (\text{outNode})$.

Thus, our task is now to find nonlinearities $F$ and $G$ so that

$a(\text{outNode}) \approx  F( G (h _K(1), …, h _K(|V|)), x )$.

Now, the set $(h _K(1), …, h _K(|V|))$ is permutation equivariant with respect to node indexing.
As suggested in [1], transformers are universal approximators of permutation equivariant functions. Therefore we use G as a transformer. Furthermore, we apply another neural network F on top of output of G and x to predict $a$(outNode). Moreover, to provide a permutation invariant order of the nodes, we use BFS sequence which is permutation invariant. Moreover, such a sequence approximately agrees with order of computation within the computation graph.

[1] Yun et al.  Are Transformers universal approximators of sequence-to-sequence functions? ArXiv, abs/1912.10077.


Further note that our Ablation study (Table 2) in the paper  and Table 1 in global-pdf show that every component in the model contributes to performance improvement as compared to several alternatives; and the total pipeline results in significant performance gain.

---

### Author Response · Authors · 2023-08-21
**Thanks to all reviewers!**

We thank all the reviewers for their constructive and insightful comments, suggestions and engagement during the discussion period which has helped us enhance the paper. We will incorporate all their comments in the final version of the paper, if it gets accepted. Among the potential changes, some key points are:

- We will use the additional page to distribute figures and tables to improve readability.
- We will bring the results on Selection-via-Proxy from the supplementary in the main paper and also add RHO-Loss. Here, we will add more illustrative results to contrast with existing works.
- We will highlight ablation study in Table 2 in the paper more prominently.
- We will add results on ImageNet as well as elaborate the overview of our method in the light of theoretical underpinning.

---

### Decision · Program_Chairs · 2023-09-21

**Decision:**

Accept (poster)

**Comment:**

This paper proposes an example selection method called SubSelNet, which generalizes subset selection to unseen models. In discussion, all the reviewers agreed towards acceptance of this paper, and I agree with this consensus. However, there were some significant structure and presentation issues, which I suggest the authors work on for their camera ready submission, taking the reviewers' feedback into account. While the empirical results and overall contributions seem to merit acceptance, this paper will be much more impactful if it is presented more clearly.